# Gut Microbiota Functional Dysbiosis Relates to Individual Diet in Subclinical Carotid Atherosclerosis

**DOI:** 10.3390/nu13020304

**Published:** 2021-01-21

**Authors:** Andrea Baragetti, Marco Severgnini, Elena Olmastroni, Carola Conca Dioguardi, Elisa Mattavelli, Andrea Angius, Luca Rotta, Javier Cibella, Giada Caredda, Clarissa Consolandi, Liliana Grigore, Fabio Pellegatta, Flavio Giavarini, Donatella Caruso, Giuseppe Danilo Norata, Alberico Luigi Catapano, Clelia Peano

**Affiliations:** 1Department of Pharmacological and Biomolecular Sciences, University of Milan, 20133 Milan, Italy; andrea.baragetti@unimi.it (A.B.); elisa.mattavelli@unimi.it (E.M.); giada.caredda@unimi.it (G.C.); flavio.giavarini@unimi.it (F.G.); donatella.caruso@unimi.it (D.C.); danilo.norata@unimi.it (G.D.N.); 2Institute of Biomedical Technologies, National Research Council, 20090 Segrate, Milan, Italy; marco.severgnini@itb.cnr.it (M.S.); clarissa.consolandi@itb.cnr.it (C.C.); 3Epidemiology and Preventive Pharmacology Service (SEFAP), Department of Pharmacological and Biomolecular Sciences, University of Milan, 20133 Milan, Italy; olmastronielena92@gmail.com; 4Institute of Genetic and Biomedical Research, UoS Milan, National Research Council, Rozzano, 20125 Milan, Italy; carola.conca_dioguardi@humanitas.it (C.C.D.); clelia.peano@humanitasresearch.it (C.P.); 5Institute of Genetic and Biomedical Research, National Research Council, 09042 Cagliari, Italy; andrea.angius@irgb.cnr.it; 6Department of Experimental Oncology, IEO, European Institute of Oncology, IRCCS, 20141 Milan, Italy; luca.rotta@ieo.it; 7Genomic Unit, IRCCS, Humanitas Clinical and Research Center, 20090 Rozzano, Milan, Italy; Huge@humanitasresearch.it; 8S.I.S.A. Center for the Study of Atherosclerosis, Bassini Hospital, 20092 Cinisello Balsamo, Milan, Italy; grigore.centroatero@gmail.com (L.G.); fabio.pellegatta@guest.unimi.it (F.P.); 9MultiMedica IRCCS, 20092 Cinisello Balsamo, Milan, Italy

**Keywords:** Atherosclerotic Cardiovascular Diseases, Gut Microbiota, next generation sequencing

## Abstract

Gut Microbiota (GM) dysbiosis associates with Atherosclerotic Cardiovascular Diseases (ACVD), but whether this also holds true in subjects without clinically manifest ACVD represents a challenge of personalized prevention. We connected exposure to diet (self-reported by food diaries) and markers of Subclinical Carotid Atherosclerosis (SCA) with individual taxonomic and functional GM profiles (from fecal metagenomic DNA) of 345 subjects without previous clinically manifest ACVD. Subjects without SCA reported consuming higher amounts of cereals, starchy vegetables, milky products, yoghurts and bakery products versus those with SCA (who reported to consume more mechanically separated meats). The variety of dietary sources significantly overlapped with the separations in GM composition between subjects without SCA and those with SCA (RV coefficient between nutrients quantities and microbial relative abundances at genus level = 0.65, *p*-value = 0.047). Additionally, specific bacterial species (*Faecalibacterium prausnitzii* in the absence of SCA and *Escherichia coli* in the presence of SCA) are directly related to over-representation of metagenomic pathways linked to different dietary sources (sulfur oxidation and starch degradation in absence of SCA, and metabolism of amino acids, syntheses of palmitate, choline, carnitines and Trimethylamine *n*-oxide in presence of SCA). These findings might contribute to hypothesize future strategies of personalized dietary intervention for primary CVD prevention setting.

## 1. Introduction

Atherosclerotic Cardiovascular Diseases (ACVD) still contribute significantly to excessive mortality, despite pharmacological weapons substantially improving their treatment [1]. Moreover, the preventive perspectives are complicated yet at early stages because the approaches to identify high risk subjects are complex. First, the presence of focal atherosclerotic lesions, detected by currently used techniques (such as ultrasound) identify subjects at increased risk of ACVD [1], although tracking Subclinical Carotid Atherosclerosis (SCA) progression by preclinical markers (including carotid Intima-Media Thickness—IMT) remains an important step to further identify subjects in primary prevention [2]. Second, IMT and the presence of focal carotid vascular lesions (a more robust indicator of ACVD risk [1]) are both predicted by classical cardiovascular risk factors (CVRFs) (e.g., Type 2 Diabetes (T2D), Metabolic Syndrome (MetS), dyslipidemia), and patterns of individual predisposition to other emerging factors such as low-grade inflammation [3,4]. Thereby, early approaches to reduce the onset and burden of these factors in a personalized fashion are surmised [5] and become feasible with the understanding of modifiable factors, like changes in lifestyle and exposure to environmental factors that differently affect risk of ACVD [6]. Under this vision, diet represents the first target for personalized approaches since digestion, the metabolism of nutrients and the absorption of potentially bioactive compounds shaping immune-metabolic functions of the host are demanded to Gut Microbiota (GM), which reflects the individual interaction with the environment [7]. Actually, the physiological cross-talk between diet with GM richness and variety (namely “eubiosis”) is well proven. On one hand, acute changes in dietary habits rapidly re-shape GM composition [8] while, on the other, GM compositions appears to be a superior factor (even beyond the genetic and clinical background of the host) in determining individual metabolic and inflammatory postprandial response to different foods [9]. This cross-talk is supported by the exposure to different dietary patterns, contributing to the absorption of a multitude of dietary metabolites that, in turn, critically foster well-described anti- or pro-inflammatory metagenomic molecular pathways. For example, some short-chain fatty acids are beneficial (e.g., butyrate and propionate activate intestinal gluconeogenesis), whereas others promote pathogenic mechanisms leading metabolic impairments (e.g., acetate promotes hepatic gluconeogenesis predisposing to glucose intolerance [10]). Bile salts contribute to preserving the intestinal barrier while favoring the proliferation of inflammatory bacteria like Clostridia by the Aryl hydrocarbon-receptor system [11,12]. Additionally, branched-chain amino acids [13] in protein-based foods and products of tyrosine/tryptophan metabolism (*p*-Cresol and indoles) exert inflammatory potential and promote insulin resistance [14]. Finally, the atherogenic properties of Trimethylamine *n*-oxide (TMAO), metabolized in the liver starting from dietary choline, have been deeply described [15,16,17,18,19,20]. Despite these robust data, whether the individual exposure to different dietary sources relates with GM taxonomic alterations (namely, “dysbiosis”) even during initial stages of atherosclerosis and before clinical establishment of ACVD is still to be understood. Whether this relation is explained by the activation of bacterial cellular pathways involved in the metabolism of dietary sources towards potentially active metagenomic compounds represents a significant add-on the potential causal effect of GM in atherosclerosis [21,22,23]. This scientific question is of current clinical concern given the continuous changes in dietary sources nowadays among societies [6]. However, methodological criticisms affect GM composition analysis [24], the design of interventional dietary trials is scarce up to now, small-sized trials gave contrasting data about the effect of dietary intervention on changes of GM composition and subsequent effect on markers of ACVD risk [25]. This sets the stage for an immediate clinical value, since the clustering of taxonomic and metagenomic signatures with individual dietary lifestyle might represent a pioneering approach of primary prevention, identifying patients among the population at increased risk of future occurrence of CVD. We here address the relation between functional metagenomic signatures and individual exposure to diet during subclinical manifestation of CVD, studying people from a general population-based study, in primary prevention, with low prevalence of CVRFs and characterized by different stages of SCA.

## 2. Materials and Methods

### 2.1. Study Population

For the purposes of this study, we collected fecal samples from 345 subjects in primary prevention for CVD from the population-based study representative of the general population of the northern area of Milan (Progressione delle Lesioni Intimali Carotidee—PLIC), which has been extensively described elsewhere [26,27]. Subjects of this study were screened at the Center for the Study of Atherosclerosis at E. Bassini Hospital (Cinisello Balsamo, Milan, Italy) for personal and familial clinical history and for absence of previous CVDs and personal history of T2D or MetS (defined according to validated criteria). Additionally, we excluded: (i) subjects reporting use of glucose-lowering drugs, (ii) with positive personal history of CVD (either ischemic heart disease, ST segment elevation or non-ST elevation myocardial infarction, aortic-coronary by-pass grafting, angioplasty, transient ischemic attack, stroke, heart failure from Class II to IV (according to New York Heart Association (NYHA) definition or documented peripheral arteriopathy), (iii) with MetS (defined according to harmonized criteria of the American Heart Association [28]), (iv) chronic kidney disease (Glomerular Filtration Rate, GFR, <60 mL/min or documented albuminuria > 30 mg/g), (iv) pregnancy and (v) reported malignancies. Data management and statistical analyses were performed with the coordination of the Epidemiology and Preventive Pharmacology Centre (SEFAP) of the University of Milan. The study was approved by the Scientific Committee of the University of Milan (SEFAP/Pr.0003). Informed consent was obtained from subjects (all over 18 years-old), in accordance with the Declaration of Helsinki. Systolic and diastolic blood pressure and Body Mass Index (BMI), waist and waist/hip ratio were measured. Information on the presence of hepatic steatosis, available on a subgroup of 133 subjects, was defined via ultrasound, as per already published protocols [26]. Blood samples were collected from the antecubital vein after 12 h fasting on NaEDTA tubes (BD Vacuette^®^, Franklin Lakes, NJ, USA) and then centrifuged at 3000 rpm for 12 min (Eppendorf 580r, Eppendorf, Hamburg, Germany) for biochemical parameters profiling including: total cholesterol, HDL-C, triglycerides, ApoB, ApoA-I, glucose, liver enzymes, creatinine and creatinine-phospho kinase (CPK). Measurements were performed using immuno-turbidimetric and enzymatic methods thorough automatic analyzers (Randox, Crumlin, UK). LDL-C was derived from Friedewald formula. Separately, whole blood in NaEDTA tubes was used for hematocrit analysis to derive a total count of leukocytes and their fractions (neutrophils, lymphocytes, monocytes, eosinophils and basophils, indicated as cells*1000/microliter).

Fecal samples of 345 subjects were collected and used for the analysis of GM taxonomic composition. SCA was defined by ultrasound-based analysis of bilateral carotid arteries as previously described [27]. In detail, common carotid IMT (one centimeter from the bulb) was measured in longitudinal view, far wall, by a high resolution B-mode ultrasound-based system (Vivid S5—GE Healthcare, Wauwatosa, WI, USA) connected to linear probe—4.0 × 13.0 MHz frequency; 14 × 48 mm footprint, 38 mm field of view). A mean value for both sides was averaged. “+ IMT” was determined as the presence of IMT above the 75th percentile of the median IMT for a Caucasian population according to ASE guidelines [29].

SCA was defined when mean IMT was ≥1.3 mm or in presence of focal atherosclerotic lesions larger than 1.3 mm using a manual caliper in longitudinal view either in far or near wall and over every carotid tract (common, bulb section, bifurcation, internal or external branches). In two scans performed by the same operator in 75 subjects, the mean difference in IMT was 0.005 ± 0.002 mm and the coefficient of variation (CV) was 1.93%. The correlation between two scans was significant (*r* = 0.96; *p* < 0.0001). The combination of information from IMT measurement and from presence/absence of SCA allowed four different SCA stages to be identified: subjects without intimal thickening and without SCA (“−IMT/−SCA”, *n* = 23); subjects with intimal thickening but without SCA (“+IMT/−SCA”, *n* = 173); subjects without intimal thickening but with SCA (“−IMT/+SCA”, *n* = 121); subjects with both intimal thickening and SCA (“+IMT/+SCA”, *n* = 23).

Whole shotgun metagenomic sequencing analyses were performed on the same fecal samples of 23 “−IMT/−SCA” and 23 “+IMT/+SCA”, whose clinical characteristics are reported in (Appendix A). Further vascular characterization according to validated criteria [30] allowed, among the +IMT/+SCA group, subjects with “no advanced SCA” (stenosis < 30% and *p*/S < 125 cm/s) vs. the “advanced SCA” (stenosis 30% and elevation of the *p*/S wave in the bilateral internal carotid branches) to be distinguished. The advanced SCA was then divided by further characterization identifying: (a) SCA causing stenosis between 30 and 50% with *p*/S < 125 cm/s; (b) SCA causing stenosis between 50 and 70% and *p*/S between 125 and 250 cm/s; (c) SCA causing stenosis over 70% and *p*/S over 250 cm/s. We evaluated echolucencies of the atherosclerotic lesions among all subjects from the −IMT/+SCA and from the +IMT/+SCA group, using grey-scale definition and parameters of the QuickScan^®^ and autoIMT^®^ software included in the ultrasound machinery (Samsung HM70a^®^, Samsung^®^, Seoul, South Korea).

Additional information, clinical criteria, and determination of biochemical parameters are reported in Appendix A.

### 2.2. Lifestyle Data, Collection and Analysis of Dietary Habits

Subjects self-reported their level and type of physical activity and smoking habit and information about individual diet were collected in the PLIC study as previously reported [31]. In detail, all subjects were requested to complete a semi-quantitative daily food diary representative of seven days before the clinical evaluation and collection of the fecal sample. The food diary was administered to subjects following instructions about the reporting of quali/quantitative dietary information by two dieticians (blinded on subject’s clinical history). In the food diary, subjects reported for each meal (breakfast, lunch, dinner and snacks) the foods, the brand names of foods (where applicable), the methods of preparation and dressings. During the seven days, dieticians were available for help and to provide more instructions to subjects by phone or by email. A portion reference from validated color photographs (the “Atlante Fotografico delle Porzioni degli Alimenti”; [32]) was also given to subjects, for further help in the interpretation of food quantities. Then, after seven days, the filled-out food diary was analyzed by dieticians during the outpatient evaluation in front of the subject: (i) to clarify details and improper indications and (ii) to derive individual daily energy and the seven-day dietary averaged nutrient intakes (as g/week), referring to the Italian BDA database [33]. BDA- Food Composition Database for Epidemiological Studies in Italy—2015). Additionally, BDA and the reference values of the Italian Society of Nutrition (“LARN”, Livelli di Assunzione di Riferimento di Nutrienti ed Energia) were used to exclude outlier data about energy intake, deriving from the improper self-reporting of the subject.

### 2.3. DNA Extraction from Fecal Samples

Total microbial DNA from all the fecal samples collected has been extracted as previously described [34], since the protocol herein described was specifically modified to allow an efficient and unbiased bacterial DNA extraction from human fecal samples. Genomic DNA quality was assessed by using the TapeStation 2200 system (Agilent, Santa Clara, CA, USA); only samples having a DNA Integrity Number (DIN) > 4 were used for successive analyses.

### 2.4. Libraries Construction and Sequencing

#### Microbiome 16S Analysis

For each sample, the V3–V4 region of the 16S rRNA gene was PCR-amplified by using primers carrying overhanging adapter sequences (primer selection originally described in [35]), following the Illumina 16S Metagenomic Sequencing Library Preparation protocol [36] (Illumina, San Diego, CA, USA), and libraries were barcoded using dual Nextera^®^ XT indexes (Illumina). Indexed libraries were pooled at equimolar concentrations and sequenced on a HiSeq 2500 Illumina sequencing platform generating 2 × 250 bp paired-end reads, according to manufacturer’s instructions (Illumina).

### 2.5. Metagenome Analysis

A total of 46 Metagenomic shotgun libraries were prepared from the DNA extracted from fecal samples from 23 “−IMT/−SCA” and 23 “+IMT/+SCA; dual indexed libraries were prepared following the Nextera^®^ DNA Flex Library Prep Kit (Illumina); then, they were pooled at equimolar concentrations and sequenced on a Novaseq 6000 Illumina sequencing platform, generating 2 × 100 bp paired-end reads.

### 2.6. Statistical and Bioinformatic Data Analysis

#### Microbiome 16S Data Analysis

The 16S rRNA raw sequences were processed through a bioinformatic pipeline composed of PANDAseq [37] and QIIME (release 1.8.0 [38]); Operational Taxonomic Units (OTUs) were assigned at 97% similarity level and classified against the Greengenes database (release 13.8; [39]). Biodiversity and distribution of the microorganisms were characterized via alpha- and beta-diversity analysis evaluating specific metrics and distances. Statistical evaluation was performed by non-parametric Monte Carlo-based tests and by analysis of variance with partitioning among sources of variation (“adonis” function) in the R package “vegan” (version 2.0–10; [40]) for alpha- and beta-diversity, respectively. Differences in abundances of bacterial taxa were analyzed by non-parametric Mann-Whitney U-test using MATLAB software (Natick, MA, USA). Unless otherwise stated, *p* < 0.05 were considered as significant for each statistical analysis. Detailed procedures are available as Appendix A.

### 2.7. Metagenome Data Analysis

Metagenomic reads were quality filtered using the recommended pipelines from the Human Microbiome Project [41,42]. Resulting reads were then processed by HUMAnN2 pipeline (v. 0.11.2 [43]). In order to compensate for different sequencing depths, all measures were expressed as copies-per-million (CPM). 

Alpha- and beta-diversity analyses were performed on species-level taxonomic classification and MetaCyc reaction-level functional classification [44], using non-phylogenetic indexes and distances in QIIME. Statistical evaluation was performed as described above. Pathways were grouped to upper levels thanks to their lineage association in MetaCyc. Detailed procedures are available as Appendix A.

### 2.8. Dietary Data Analysis

Statistical data of nutrients composition for individuals with and without SCA was performed by employing the non-parametric Mann-Whitney U-test. Overall separation between patients was assessed calculating Bray-Curtis distances among patients on the basis of the nutrients table and “adonis” function in the R package “vegan” was used. In order to assess the correlation between dietary and microbial composition data, the RV coefficient [45] was calculated; coefficient statistical significance of the coefficient was calculated by 99,999 random permutations [46].

### 2.9. Data Availability

Sequencing data of 16S rRNA amplicons (raw reads, *n* = 345) and metagenomes (after removal of human sequences and duplicates, *n* = 46) have been deposited in NCBI Short-Read Archive (SRA) under accession number PRJNA615842 [47].

## 3. Results

### 3.1. Gut Microbiota Dysbiosis Associates with Cubclinical Carotid Atherosclerosis

We identified 144 subjects with SCA by carotid ultrasound examination and 201 gender-matched subjects without SCA (clinical characteristics of both groups reported in Table 1).

Subjects with SCA were older than those without SCA, showed higher waist/hip ratio (0.91 ± 0.08 vs. 0.87 ± 0.08, *p* < 0.001) and they were more hypertensive (on anti-hypertensive 56.55% vs. 37.24% respectively, *p* = 0.004). LDL-C was 7 mg/dL lower in subjects with SCA vs. those without (114.90 ± 27.83 vs. 121.20 ± 27.23 mg/dL, *p* = 0.037) because of the higher prevalence of lipid lowering treatments (58.62% vs. 40.31% respectively, *p* =0.008). Subjects with SCA presented with comparable plasma high-sensitivity C-Reactive Protein (hs-CRP) but with higher blood monocytes counts versus those without SCA (0.11 (0.06–0.21) mg/dL vs. 0.10 (0.05–0.19) mg/dL, *p* = 0.179 for hs-CRP; 0.53 ± 0.15 vs. 0.49 ± 0.14 ∗ 1000 cells/µL, *p* = 0.019 for monocytes).

We firstly investigated whether changes in GM composition still occur over SCA progression, thereby analyzing taxonomic GM composition of the entire cohort (*n* = 345) by 16S rRNA-based sequencing (Appendix A), and we then performed a metagenome shotgun sequencing on a subset of subjects with +IMT/ + SCA and of those with –IMT/-SCA phenotype (*n* = 46), selected according to SCA presence and IMT measurements as described above. This dual strategy allowed to gather, with a high degree of consistency (Appendix A), information about different relative abundances of bacterial genera and species in the presence or absence of SCA.

The GM taxonomic composition significantly differed between subjects with SCA (*n* = 144) and those without (*n* = 201) (*p* = 0.016, unweighted Unifrac distance, Figure 1A), although no changes in GM richness were found (*p* > 0.05 in all alpha-diversity metrics). The 16S rRNA-based analysis highlighted increased relative abundance of members of Escherichia (2.8% vs. 1.4%, *p* = 0.008 in SCA and no SCA subjects, respectively) and Oscillospira (6.5% vs. 5.7%, *p* = 0.013 in SCA and no SCA subjects, respectively) genera (Figure 1B,C, Appendix A) in subjects with SCA.

Notably, these differences in GM diversity in the presence of SCA were significantly explained by those from the subset of 23 subjects with +IMT/ +SCA phenotype, analyzed by metagenome shotgun sequencing, as compared to all the other three groups (*p* ≤ 0.030 and *p* ≤ 0.004 for all pair-wise comparisons on unweighted and weighted Unifrac-based PCoA, respectively, Appendix A). Moreover, we found few genera whose taxonomic abundances were significantly different in fecal samples from the subset of subjects with +IMT/+SCA (*n* = 23). In particular, Escherichia, Shigella and Streptococcus were those mostly significantly increased while Bacteroides were reduced in subjects with the most advanced SCA stage (Appendix A).

On top of significantly different metagenomic profiles in subjects with −IMT/−SCA vs. +IMT/SCA, both on alpha (*p* = 0.001, permutation-based *t*-test on observed species metrics) and beta-diversity (*p* = 0.002, adonis test on Bray-Curtis distance) measures, this strategy allowed the identification of increased abundance of *E. coli*, as well as of members of the Streptococcus genus (i.e., *S. salivarius*, *S. parasanguinis*, *S. anginosus*) in metagenomes of subjects with +IMT/+SCA. Vice versa, we found increased abundance of members of Bacteroides genus (i.e., *B. uniformis* and *B. thetaiotaomicron*) in the metagenomes of subjects with −IMT/−SCA (Table 2). Together these data allowed to conclude that specific taxonomic and metagenomic markers can be found still during early stages of carotid intimal thickening and SCA.

### 3.2. Functional Relevance of GM Dysbiosis over Subclinical Carotid Atherosclerosis

In order to highlight whether different shapes in GM composition over SCA stages have a functional relevance, we harnessed data from metagenome shotgun sequencing to predict the MetaCyc reactions (see Appendix A) mostly encoded in GM of subjects with +IMT/+SCA (*n* = 23).

Notably, in the metagenomes of this group of subjects, we predicted a higher number of MetaCyc reactions as compared to those without subclinical atherosclerosis (−IMT/−SCA, *n* = 23) (*p* = 0.009 and *p* = 0.003 for observed species and Simpson’s index, respectively) (Figure 2A); moreover, considering MetaCyc pathway abundances, significantly different functional profiles were evidenced (*p* = 0.001, adonis test on Bray-Curtis distances) (Figure 2B).

Based on these findings we then sought to identify which were the over- or down-represented bacterial metagenomic pathways in GM of subjects with +IMT/+SCA versus those without subclinical atherosclerosis (−IMT/−SCA). Moreover, the contribution of each species to MetaCyc reaction pathways was elucidated, highlighting the bacterial species in GM related to over or down-represented pathways (Figure 3).

We found that *E. coli* was the bacterial species associated with altered microbial pathways in samples from +IMT/+SCA individuals, including an increase in those related to the biosynthesis of palmitate, arginine, glutamine, biotin, phylloquinone, ubiquinone, menaquinone and phosphatidylethanolamine (PE). Notably, these pathways all clustered together in metagenomes from +IMT/+SCA subjects, characterized by advanced SCA stage. 

By contrast, *Faecalibacterium prausnitzii* was found to contribute to thirteen significantly different MetaCyc reactions (with sulfur oxidation, starch degradation and multiple biosynthetic routes of purine and pyrimidines as the most over-represented) in samples from −IMT/−SCA subjects.

Together, these observations allowed the conclusion that taxonomic changes in GM composition, occurring during first stages of subclinical atherosclerosis and coinciding with individual exposure to dietary sources, highlight functional metagenomic relevance.

### 3.3. Individual Diet Clusters with Changes in Taxonomic GM Composition and SCA

These observations supported that early re-shape in taxonomic GM composition occurs during the first SCA stages and prompted us to explore whether this might be associated with different individual exposure to dietary sources.

Daily alimentary habits were profiled by the analysis of daily food diaries, filled in by subjects one week before the collection of the fecal sample (see Appendix A).

Subjects with SCA reported to consume higher amounts of mechanically separated meats (*p* = 0.013) although a similar amount of ham, salami, sausages (*p* = 0.414), meat products and substitutes (*p* = 0.470) and a similar amount of not preserved meat (beef, veal, poultry and pork) (*p* = 0.683) as compared to subjects without SCA. Moreover, they reported consuming more dried fruits (*p* = 0.02), also with a trend towards higher quantities of fruits (both processed and fresh) and eggs. By contrast, subjects without SCA reported to consume a higher amount of cereals (*p* = 0.009), starchy vegetables (*p* = 0.027), milky products and beverages (*p* = 0.004 and *p* = 0.016, respectively), yoghurts (*p* = 0.047) and bakery (*p* < 0.001) as compared to those with SCA (Figure 4A).

Moreover, these dietary intakes showed more significant correlations to bacterial genera in subjects with SCA as compared to those without SCA (Figure 4B; red square represent positive correlation while blue square indicates negative correlation; yellow dots indicate significant correlations, *p* < 0.05). Of note, some of these correlation recapitulated data from metagenomic analysis. For example, increased relative abundance of *Escherichia* was inversely related with intakes of yoghurts, pulses and fresh vegetables while positively correlated to bakery only in individuals with SCA.

We were overall able to cluster intakes of specific food patterns (including milk-derived products, yogurts, total and leafy vegetables, processed cereals, mechanically separated meats, fish, bread and bakery products) that significantly co-segregated with the taxonomic diversities between subjects with SCA as compared to subjects without SCA (RV coefficient between nutrients quantities and microbial relative abundances at genus level = 0.65, *p*-value = 0.047) (Figure 4C). In addition to this finding, we also found similarly significant changes (*p* = 0.04) in the individual diet in subjects with +IMT/+SCA versus those without subclinical atherosclerosis (−IMT/−SCA) (not shown), further supporting the interaction of dietary exposure on GM changes and early stages of SCA.

Together these data support that changes in taxonomic GM composition, occurring still at subclinical stages of SCA, cluster and occur together with differences in individual exposure to dietary sources.

## 4. Discussion

Despite effective strategies for the treatment of patients with clinically manifest ACVD [1], the identification of subjects at higher risk of future development of the disease is currently far from optimal, due to hardwired CVRFs [1] and individual predisposition to low-grade inflammation [3]. Therefore, personalized approaches are sought, targeting the host-derived CVRFs and those dependent on the individual exposure to environment, in a tailored way [6]. The microbiome appears the most relevant potential target under this perspective, due to its involvement in the metabolism of dietary sources and since it is particularly sensitive to rapid changes in dietary habits [7]. We hypothesized, via an innovative research design, that GM changes in taxonomic and functional signatures still occur during subclinical stages of ACVD, before the clinical manifestation of CVRFs.

Currently, our data add further knowledge about the relation between GM and ACVD since, in comparison to other studies involving patients with clinically established ACVD, (either as coronary ischemic atherosclerosis [23], cerebrovascular events [23,40], ST-elevation myocardial infarction (STEMI) [21], stable angina or coronary artery disease [22]), we here show changes in taxonomic GM composition in subclinical stages of ACVD progression, when the effect of high cardio-metabolic impairment is not yet clinically evident.

We analyzed the GM profile in 345 subjects (the majority of whom were aged between 60 and 80 years-old), prevalently lean/overweight, without clinically manifest ACVD, T2D and MetS. Taxonomic compositions taken from this cohort (one of the largest of this kind) were comparable to those found in other geographically distant Italian cohorts [48,49,50] although they did not confirm some correlations between relative abundances of genera/species and cardio-metabolic markers (Appendix A), that were previously reported in populations characterized by different clinical phenotypes. For example, we did not find correlation between relative abundances of *Clostridium* species with increased BMI and with higher fasting glucose, findings that were reported in severe obese post-menopausal women (34.5 Kg/m^2^ as mean BMI) [51], in 70 year-old overweight subjects (28.0 Kg/m^2^ as mean BMI) but with T2D [52] and in younger and prevalently lean subjects (43–63 y-old and 23.7 Kg/m^2^ as mean BMI) with diabetes (possibly including type 1 forms) [53]. Additionally, hepatic steatosis (which was only ultrasound-determined and poorly prevalent in our cohort) as well as plasma hs-CRP, a marker of low-grade inflammation, did not correlate either with GM diversity (while recently reported in morbidly obese patients [54] and in liver steatosis [55]), or with a reduced relative abundance of specific bacterial species (e.g., *Akkermansia muciniphila* (less abundant in insulin resistance [56] and in obese individuals [57]). Although these data might exclude an impact of a gut-liver connection during early stages of SCA, in depth evaluation on larger cohorts with more advanced stages of liver disease should is warranted.

Vice versa, we found significant relations with taxonomic species which down- or over-represent metabolic pathways of multiple dietary sources, hereby supporting an early immune-inflammatory activation of GM dysbiosis engaged by different dietary sources over SCA stages, before the clinical manifestation of ACVD and CVRFs. Different lines of evidence sustain our scientific question. Firstly, in contrast to previous reports in patients with metabolic syndrome [54,58], we here found an inverse relation between reduced HDL-C and increased abundance of Escherichia only in subjects with +IMT/ +SCA, who also showed reduced HDL-C and increased plasma levels of the atherogenic molecule TMAO (a coincidence with previous data [15,16,17,18,19,20,59,60], see Appendix A). Whether this is a consequence of TMAO produced by Escherichia (which has been reported to inhibit both HDL-mediated reverse cholesterol transport and intestinal HDL lipoprotein maturation [61]) requires further investigation. Additionally, we observed that *E. coli* caiTABCDE operon genes (encoding for membrane transport and metabolism of L-Carnitine to γBB and TMA [15,19]) were overrepresented in subjects with +IMT/ +SCA (see Appendix A). Carnevale et al. [62] recently showed higher *E. coli* abundance in GM of patients with STEMI, correlating to increased systemic Lipopolysaccharide (LPS) absorption and infiltration in atheromas from endarterectomies leading to macrophage activation [48]. In our investigation, we did not find increased plasma levels of zonulin in +IMT/+SCA subjects (see Appendix A), therefore prompting the exclusion of systemic LPS absorption through a “leaky-gut” [63] yet at initial stages of atherosclerosis. Vice versa, we found a more over-expressed biosynthetic pathway of PE (strictly linked to the hepatic conversion to atherogenic TMAO [15] mediated by the L-carnitine/γBB/TMA metabolic cascade [64]), as documented by the data presented in Appendix A, together with an increased number of circulating monocytes and neutrophils in subjects with SCA. These subjects reported indeed to consume only higher amounts of mechanically separated meats but a similar amount of unprocessed meats (as more complex food matrices containing other nutrients, phospholipids and probiotics that have been not associated with higher ACVD risk [65]) versus subjects without SCA. Future analyses and interventional dietary approaches are requested to unveil whether these connections reflect a gut-bone marrow connection fostering an activation of the innate immune system.

Secondly, we found in metagenomes of +IMT/+SCA subjects a reduced contribution of pathways (such as starch degradation, sulfur oxidation and the biosynthetic routes of purine and pyrimidines) encoded by *Faecalibacterium prausnitzii*, previously reported to be actively involved in gut permeability through the production of anti-inflammatory butyrate [66,67,68]. This finding coincides with intakes of fibers, carbohydrates and proteins from higher amounts of starchy vegetables, milky products and beverages, yoghurts and bakery products that subjects without SCA reported consuming. In fact, inulin-type fructans, fructo-oligosaccharides, polydextrose or soluble corn fiber support the proliferation of *Faecalibacterium prausnitzii* which, in turn, mediates the metabolism of fibers [68], ensures gut physiological transit time [69] and attenuates the pro-inflammatory potential of specific dietary proteins [70].

We have to acknowledge several limitations in our study. Firstly, this epidemiological study, gathering self-reported dietary data, does not allow the unveiling of the actual relation of causality. These limitations pave the road to dissecting this aspect in the near future, a perspective that might be pursued: (i) by clustering a larger number of subjects on the basis of their exposure to different dietary patterns (daily collected using smartphone health apps or other health mobile technologies to improve self-reporting) or (ii) by longitudinally evaluating the actual effect of a single dietary pattern/habit on changes of GM composition/functionality in subjects with different SCA stages. Different hurdles undermining these approaches are to be accounted in the design of lifestyle and dietary intervention studies (e.g., technical criticisms, difficulties in the interpretation of their resulting data and the export to wider populations [24]). The cross-sectional design of this investigation cannot rule out the burden of CVRFs prior to the examination, although significant differences we observed in microbial composition comparing beta-diversity measures via Principal Coordinate Analysis (PCoA) still take into account potential confounders, since they are based on multivariate models, associated with data-reduction techniques that produce a set of uncorrelated (orthogonal) axes to summarize the variability in the data set. Although the effect of pharmacological interventions over time (especially anti-hypertensive and lipid lowering drugs) cannot be ruled out, we did not find differences in the GM alpha and beta diversities when comparing subjects with pharmacological therapy and those with any treatments [23,71]).

Secondly, subjects with SCA were older in our cohort and we acknowledge that age, the principal predictor of faster SCA progression over time [4], might act as a confounding factor in the relation between GM taxonomy and SCA. However, the majority of subjects in our study were aged between 60 and 80 y/old (Table 1), therefore prompting us to conclude that data from younger cohorts or longitudinal evaluation on this same cohort are needed in order to dissect how much age interacts in the relation between GM composition and SCA progression over time.

Thirdly, our knowledge of the actual differences in metagenome-encoded functions and their relation to the host health is limited by the fact that a relevant part of the metagenome still remains undescribed (e.g., in our data, about half of the metagenomic reads could not be mapped and a majority of the mapped ones could not be annotated to a known reaction or pathway), due to the lack of complete description of bacterial genes in databases.

Finally, we did not perform a complete analysis of circulating proteomic and metabolomics markers validating the metagenomics pathways that emerged from our data. Since multiple circulating proteins and metabolites (the majority of which are of lipid origin) have been recently associated with different types of dietary patterns [72], this aspect is of relevance and will be analyzed in the near future.

However, to the best of our knowledge, this is the first extensive taxonomic and metagenomic characterization connecting GM dysbiosis with individual diet and SCA. These cross-sectional data aim at setting the stage for future longitudinal studies and dietary interventions, testing if personalized modifications in dietary habits over time could affect GM composition contributing to the prevention of the onset of CVRF and the clinical manifestation of ACVD.

## Figures and Tables

**Figure 1 nutrients-13-00304-f001:**
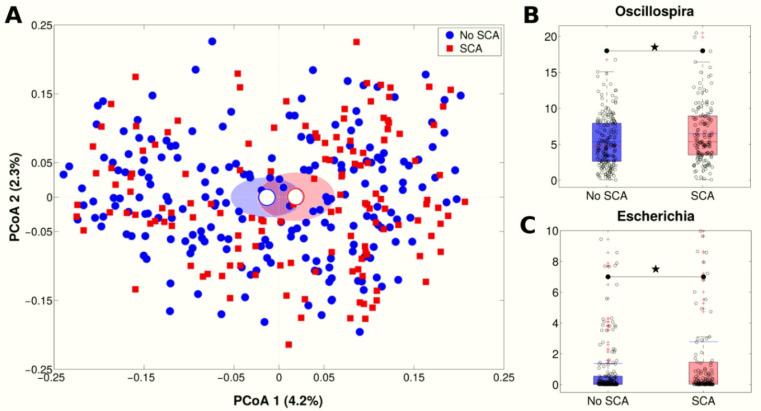
Taxonomic Gut Microbiota (GM) differences over SCA progression. (**A**) Principal Coordinate Analysis (PCoA) plot of the unweighted Unifrac distances; data were divided according to experimental category (SCA vs. no SCA); each point represents a sample; centroids are calculated as the mean coordinate of all samples per experimental category; ellipses represent the standard error of the mean (SEM)-based estimation of the variance. The first and second components of the variance are shown. (**B**,**C**) Boxplots of relative abundances of (**B**) Oscillospira and (**C**) Escherichia genera, according to SCA vs. no SCA experimental categories. Red lines indicate median and blue lines indicate mean values. Star indicates a significant difference (*p* < 0.05, Mann-Whitney U-test).

**Figure 2 nutrients-13-00304-f002:**
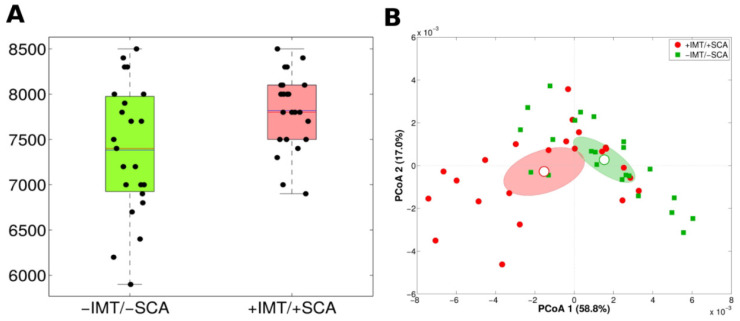
Hit functional markers associated with +IMT/+SCA. (**A**) Boxplots deriving from level L4 of the MetaCyc pathway hierarchy based on shotgun metagenome sequencing of +IMT/+SCA (*n* = 23) and −IMT/−SCA (*n* = 23) individuals as determined by “observed_species” metric. (**B**) Principal Coordinates Analysis (PCoA) plots of Bray-Curtis distances among +IMT/+SCA and −IMT/−SCA samples calculated on level L4 functional MetaCyc pathway classification. Each point represents a sample; centroids are calculated as the mean coordinate of all samples per experimental category; ellipses represent the standard error of the mean (SEM)-based estimation of the variance. The first and second components of the variance are shown.

**Figure 3 nutrients-13-00304-f003:**
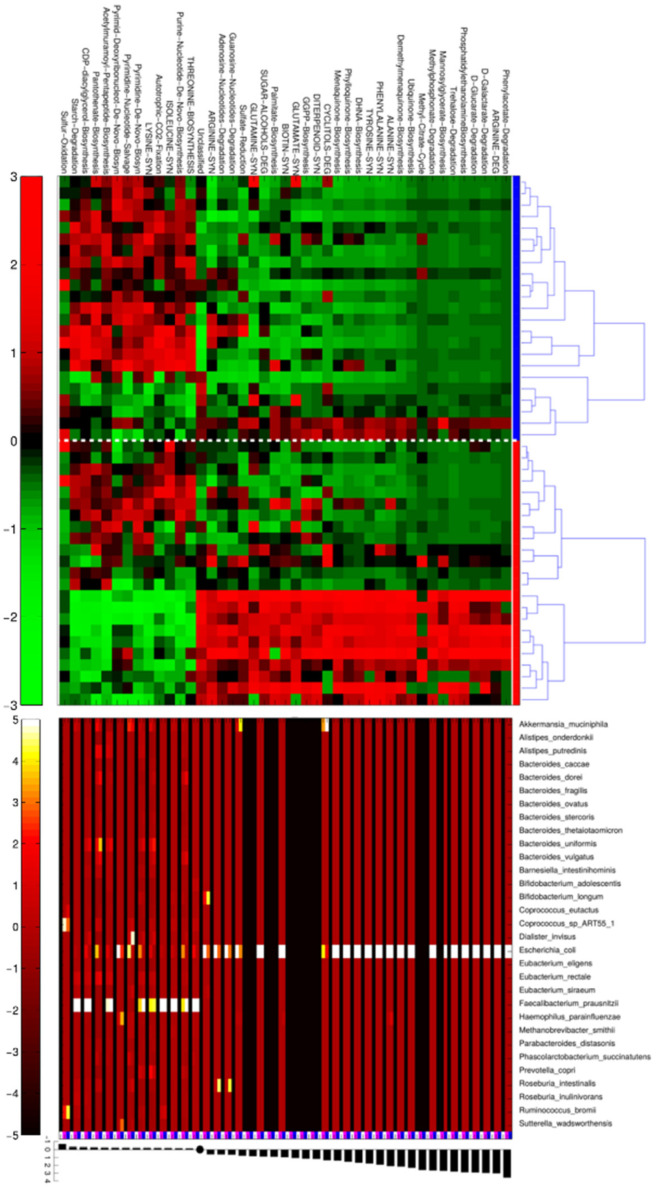
Metagenomic pathways and bacterial genera more expressed with +IMT/+SCA. Heatmap on top shows the MetaCyc L4-pathways statistically different between +IMT/+SCA (*n* = 23, red bars at right) and −IMT/−SCA (*n* = 23, blue bar at right) individuals. Normalized and scaled read counts (CPM) per pathway are standardized along rows. Within each experimental group, samples are clustered for similarity using Pearson’s correlation metric and average linkage; the plot in the middle represents the relative contribution of bacterial species to each differential pathway. Values are average CPM calculated for −IMT/−SCA samples (indicated by the blue square below) and +IMT/+SCA (indicated by the magenta square below). Average CPM values are standardized on a per-column basis (i.e., on pathways), in order to highlight, for each pathway which bacterial taxa contributes most. On the bottom, of the figure, the barplot indicates log2 fold-change between +IMT/+SCA and −IMT/−SCA average CPM.

**Figure 4 nutrients-13-00304-f004:**
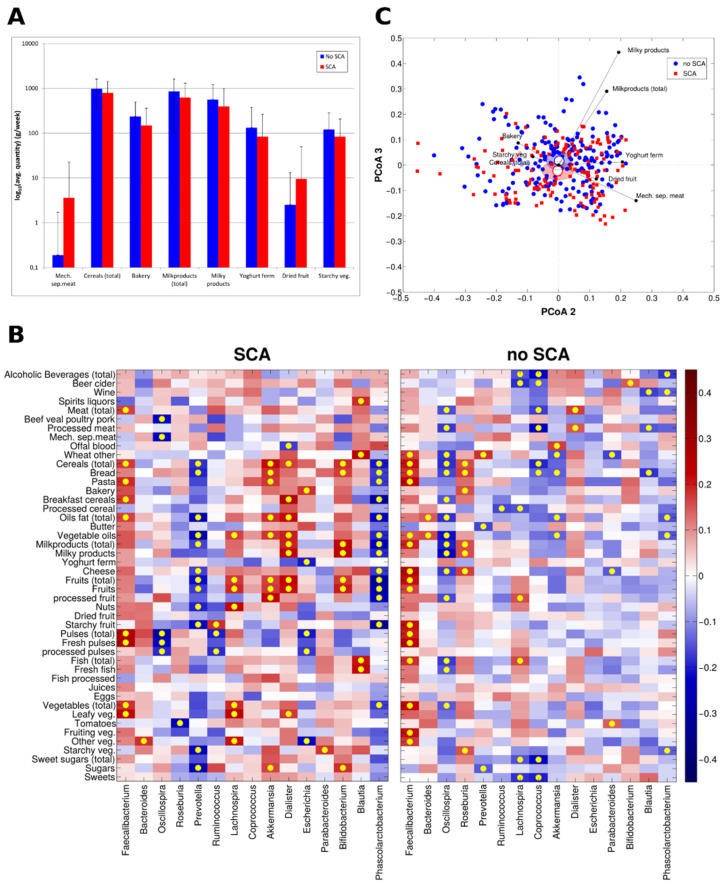
(**A**) Barplots of significantly different (*p* < 0.05, Mann-Whitney U-test) nutrients based on the analysis of daily food diaries for individuals with (*n* = 144) and without (*n* = 201) SCA. Bars represent average intake and standard deviations are represented as error bars. Due to graphical reasons, intakes were represented as log_10_ (**B**) Heatmap of the Spearman’s correlation coefficients between bacterial genera relative abundances and nutrients intake. Correlations were calculated for individuals belonging to “SCA” (*n* = 144) and “no SCA” (*n* = 201), respectively. Yellow dots correspond to significant correlation (*p*-value of the linear model < 0.05) and bacteria with an average relative abundance ≥ 1% in either experimental group were represented. (**C**) Biplot representing samples according to dietary patterns for individuals with (*n* = 144) and without (*n* = 201) SCA. Principal Coordinate Analysis (PCoA) was based on Bray-Curtis distances. Each point represents a sample, centroids represent the average coordinate for the data points in each category and ellipses represent the 95% Standard error of the mean (SEM)-based confidence interval of the data points. The second and third principal coordinates are represented. The average coordinates of the statistically different dietary component (*p* < 0.05) weighted by the corresponding abundance per sample was superimposed on the PCoA plot to identify those mainly contributing to the ordination space (black arrows).

**Table 1 nutrients-13-00304-t001:** Descriptives of the population according to SCA.

	Total	SCA		
Variable	*n* = 345	No (*n* = 201)	Yes (*n* = 144)	*p*-Value
Men gender, *n* (%)	158 (45.80)	93 (46.50)	65 (44.83)	0.760
Age (years)	67.32 (11.0)	63.83 (11.52)	72.04 (8.18)	<0.001 ***
<= 60 years-old	86	73	13	
60–70 years-old	105	64	41	
70–80 years-old	130	57	73	
>80 years-old	24	6	18	
Alcohol consumption, *n* (%)	231 (67.74)	129 (65.82)	102 (70.34)	0.380
Smoking, *n* (%)	37 (10.85)	22 (11.22)	15 (10.34)	0.800
Physical Activity, *n* (%)	175 (51.47)	115 (58.97)	60 (41.38)	0.001 ***
BMI (Kg/m^2^)	26.44 (3.90)	26.22 (3.85)	26.75 (4.00)	0.230
Lean, *n* (%)	114	76	38	
Overweight, *n* (%)	165	86	79	
Obese, *n* (%)	53	30	23	
Waist-hip Ratio	0.89 (0.08)	0.87 (0.08)	0.91 (0.08)	<0.001 ***
Systolic pressure (mmHg)	126.91 (14.41)	125.20 (13.74)	129.30 (15.01)	0.008 **
Diastolic pressure (mmHg)	75.75 (9.06)	75.06 (8.74)	76.71 (9.44)	0.100
Antihypertensive drugs, *n* (%)	155 (45.45)	73 (37.24)	82 (56.55)	0.004 ***
Total Cholesterol (mg/dL)	199.0 (32.71)	202.90 (32.13)	193.60 (32.86)	0.009 **
HDL-C (mg/dL)	60.57 (13.60)	62.23 (13.59)	58.29 (13.32)	0.008 **
LDL-C (mg/dL)	118.53 (27.62)	121.20 (27.23)	114.90 (27.83)	0.037 *
Triglycerides (mg/dL)	99.46 (37.75)	97.42 (36.76)	102.30 (39.03)	0.240
Apolipoprotein A1 (mg/dL)	157.91 (19.87)	160.30 (20.24)	154.60 (18.92)	0.008 **
Apolipoprotein B (mg/dL)	101.55 (24.70)	103.60 (25.35)	98.65 (23.57)	0.060
Lipid lowering drugs, *n* (%)	164 (48.09)	79 (40.31)	85 (58.62)	0.008 **
Fasting glucose (mg/dL)	100.36 (10.49)	100.30 (9.68)	100.40 (11.56)	0.910
Uric acid (mg/dL)	5.17 (1.44)	4.96 (1.35)	5.45 (1.52)	0.002 **
Creatinine (mg/dL)	0.84 (0.19)	0.82 (0.18)	0.86 (0.20)	0.030 *
ALT (UI/L)	21.13 (14.59)	21.63 (16.68)	20.43 (11.10)	0.420
AST (UI/L)	23.64 (6.03)	23.68 (6.39)	23.57 (5.52)	0.870
GGT (UI/L)	27.73 (36.43)	27.62 (37.85)	27.87 (34.50)	0.950
Liver steatosis, *n* (%)	33 (24.80)	11 (5.50)	20 (13.80)	0.061
CPK (mg/dL)	120.79 (62.65)	121.00 (66.09)	120.50 (57.78)	0.940
Hs-CRP (mg/dL)	0.11 (0.06–0.21)	0.10 (0.05–0.19)	0.11 (0.06–0.21)	0.176
Neutrophils (cells*10^3^/µL)	3.50 (1.25)	3.42 (1.22)	3.61 (1.28)	0.180
Leucocytes (cells*10^3^/µL)	6.26 (1.68)	6.12 (1.49)	6.46 (1.89)	0.080
Lymphocytes (cells*10^3^/µL)	2.05 (0.92)	2.00 (0.52)	2.11 (1.28)	0.370
Monocytes (cells*10^3^/µL)	0.51 (0.14)	0.49 (0.14)	0.53 (0.15)	0.019 *
Eosinophils (cells*10^3^/µL)	0.16 (0.11)	0.16 (0.10)	0.17 (0.11)	0.460
Basophils (cells*10^3^/µL)	0.04 (0.02)	0.04 (0.02)	0.04 (0.02)	0.170
C-IMT (mm)	0.77 (0.18)	0.71 (0.13)	0.85 (0.19)	<0.001 ***

Clinical parameters for the 345 selected subjects from the entire PLIC cohort (second column from left) and when divided for subjects without Subclinical Carotid Atherosclerosis (SCA) (third column from left) vs. those with SCA (fourth column from left). “***” indicates *p* < 0.005; “**” indicates *p* < 0.01; “*” indicates *p* <0.05. *p* refers to that of the two-sided Mann-Whitney U-test between subjects with and without SCA. Data are presented as mean (standard deviation) if normally distributed or as median (Inter-Quartile Range) if not normally distributed (Shapiro-Wilk test). BMI: “Body Mass Index”; HDL-C: “High Density Lipoprotein cholesterol”; LDL-C “Low Density Lipoprotein cholesterol”; ALT: “Alanine aminotransferase”; AST: “Aspartate aminotransferase”; GGT: “Gamma-glutamyl transpeptidase”; CPK: “Creatine phosphokinase”; c-IMT: “carotid Intima Media Thickness” (as averaged value of IMT values of right and left carotid arteries at the common tract). Information on ultrasound hepatic steatosis was available on 133 subjects out of total studied cohort.

**Table 2 nutrients-13-00304-t002:** Reduced and increased abundances of genera and species in +IMT/ + SCA.

Genus	Species	−IMT/−SCA	+IMT/+SCA
Escherichia	*E.coli*	0.83	7.50 (**)
	*Uncl. Escherichia*	0.18	1.93 (**)
Streptococcus	*S. salivarius*	0.23	0.50 (*)
	*S. parasanguinis*	0.06	0.44 (**)
	*S. anginosus*	0.00	0.04 (**)
Ruminococcus	*R. obeum*	0.25	0.49 (*)
Lactobacillus	*L. gasseri*	0.00	0.11 (*)
	*L. fermentum*	0.00	0.02 (*)
Dorea	*D. longicatena*	0.16	0.50 (**)
-	*C2likevirus*	0.00	0.04 (*)
Coprococcus	*Co. comes*	0.18	0.40 (**)
Clostridium	*C. leptum*	0.05	0.31 (*)
Parabacteroides	*Pa. goldsteinii*	0.06	0.30 (*)
Eubacterium	*Eu. ramulus*	0.08	0.26 (*)
Bifidobacterium	*B. dentium*	0.05	0.16 (*)
Bacteroides	*B. uniformis*	5.19	1.69 (*)
	*B. thetaiotaomicron*	0.86	0.25 (*)
Ruminococcus	*R. bromii*	2.08	1.10 (*)

List of bacterial species whose relative abundance was statistically different between +IMT/+SCA and −IMT/−SCA individuals (*n* = 23, each). “**” indicates *p* < 0.01; “*” indicates *p* < 0.05. *p* values refer to that of the two-sided Mann-Whitney U-test. For clarity, bacteria are grouped according to increase/decrease status and genus.

## Data Availability

Publicly available datasets were analyzed in this study. This data can be found in NCBI Short-Read Archive (SRA) under accession number PRJNA615842.

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
