# Peer review of "Gut Microbiota Functional Dysbiosis Relates to Individual Diet in Subclinical Carotid Atherosclerosis"

_nutrients, 2021, doi:10.3390/nu13020304_

Round 1

Reviewer 1 Report

The study by Andrea Baragetti and co-authors investigates how gut microbiota dysbiosis relates to individual diet in subclinical carotid atherosclerosis. This is based on the study of 345 subjects without previous clinically manifest ACVD and the analysis of their GM composition. Moreover, the authors examine individual diet patterns with changes in taxonomic GM composition and subclinical carotid atherosclerosis. This is a detailed study, with interesting findings but I feel that it would benefit from some clarifications and modifications.

Specific comments:

1) The 345 subjects used in the study, were divided to subjects with or without SCA (Table 1). These were then subdivided (but the rationale behind this was not explained) into +IMT/+SCA, –IMT/+SCA, +IMT/-SCA and –IMT/-SCA subgroups. While the authors are mainly describing findings from the comparisons of +IMT/+SCA and –IMT/-SCA, there are various points in the manuscript where analysis from the 4 subgroups is also described making it rather confusing as to which samples are actually being examined. If there is a reason for presenting findings from the different categories then please clearly explain in the text.

2) From the data included in Table 1, it appears that there are differences in subjects with or without SCA in several parameters, including Hepatic steatosis, uricemia, serum creatine, alcohol consumption, physical activity, lipid lowering and antihypertensive treatments. Would one not expect that changes in such parameters may also underlie some of the observed functional and taxonomic GM alterations?

3) In Table 1, please clarify which comparisons were made for statistical analysis (p values).   

4) Use of the term SCA progression may be slightly misleading as the authors are not analyzing samples of the same patient at different time points. Maybe the used of another term, eg. SCA stages, could be more appropriate.  

5) Based on the coloring scheme used in the heat map of Fig4, there doesn’t seem to be any major change in dietary habits between no SCA and SCA groups. This is in contrast to the description of the authors finding regarding significant changes in dietary habits (page 11, lines 289-295). Also, please revise the figure so that the label ‘avg_No SCA’ is shown a single line as in its current state this labeling is misleading by implying that there are 2 samples.

6) According to information on Supplementary Table 4, the SCA group shows significantly increased consumption of ‘’mechanically separated meats’’, but no statistical difference in the consumption of ‘’Meat, meat products and substitutes’’ or ‘’Beef, veal, poultry and pork, not preserved’’. This is in contrast to what is mentioned in lines 289-290 of page 11 ‘’Subjects with SCA reported to consume higher amounts of mechanically separated meats (p=0.016) and beef, veal, poultry and pork’’. Please correct text accordingly.

7) In Supplemental Figure 4 there is no data on increased plasma levels of zonulin in +IMT/+SCA subjects, as mentioned in the Discussion page 14, line 407. Instead this figure shows data on E. coli metagenetic markers associated with advanced SCA progression. Please correct.   

8) In Figure 5, the MetaCyc L4-pathways and the bacterial species of the two heat-maps are too small to read. Given that this is important information, the authors should consider increasing the font or showing only some of the most significant changes. Findings shown in this figure are important but currently they are not visible.

9) Do the authors have any other experimental evidence on bacterial metabolites (eg from serum) to confirm the observed alterations in metagenomic pathways?

10) According to this study, taxonomic changes in GM composition were shown to occur during first stages of subclinical atherosclerosis. The authors conclude that this coincides with individual exposure to dietary sources, highlighting functional metagenomic relevance. However, it is still unclear to me how GM composition is correlated with dietary patterns. Please explain in section 3.2 the analysis performed for this and describe in more detail the findings. This is an important point of this manuscript but it is not adequately explained. A figure for this should also be included.

11) Along these lines, would it not have been more appropriate if all subjects had been under similar dietary habits? Would this not represent a more homogeneous and potentially better strategy to study the functional role of the different GM composition in SCA progression?

12) Given that the study includes individuals of subclinical stages of ACVD, what do the authors believe is the most significant change observed at this stage of disease in relation to GM composition, particularly dietary pattern, or specific metagenomic pathway alteration? Do the authors envision that reversal of such change may result in prevention of disease progression?

13) Since there is a lot of bionformatical analysis and graphs included in the article, please ensure to clearly explain what samples were analyzed in each graph, why this was done and what is the main findings from each analysis. For example, there are several PCA plots included but it is not always clear as to what each one of them is showing.

14) There appears to be a lot of information (methods and results) as supplementary. This makes it very disruptive to the reader, by constantly having to refer to supplementary material, but also difficult to follow due to the large amount of information included. It may be best to condense the supplementary material and include some of the most important information in the main text. 

Author Response

Dear Reviewer#1,

Thank you very much for your helpful comments, which significantly contribute to the improvement of the manuscript. We have revised the manuscript according to your suggestions and those from the other reviewers.

Please, find below the responses to each of your comments:

The study by Andrea Baragetti and co-authors investigates how gut microbiota dysbiosis relates to individual diet in subclinical carotid atherosclerosis. This is based on the study of 345 subjects without previous clinically manifest ACVD and the analysis of their GM composition. Moreover, the authors examine individual diet patterns with changes in taxonomic GM composition and subclinical carotid atherosclerosis. This is a detailed study, with interesting findings but I feel that it would benefit from some clarifications and modifications.

Specific comments:

1) The 345 subjects used in the study, were divided to subjects with or without SCA (Table 1). These were then subdivided (but the rationale behind this was not explained) into +IMT/+SCA, –IMT/+SCA, +IMT/-SCA and –IMT/-SCA subgroups. While the authors are mainly describing findings from the comparisons of +IMT/+SCA and –IMT/-SCA, there are various points in the manuscript where analysis from the 4 subgroups is also described making it rather confusing as to which samples are actually being examined. If there is a reason for presenting findings from the different categories then please clearly explain in the text.

We thank the reviewer for this important aspect.

Carotid Intima-Media Thickness (IMT) is a marker of pre-clinical atherosclerosis although international guidelines (See Ref#1) support its measurement secondly to the detection of focal atherosclerotic lesions (SCA) to identify subjects at higher ACVD risk.

Given differences that emerged significant both for GM taxonomic diversities and exposure to dietary patterns when subjects were stratified for SCA even without considering IMT, we now set all the analyses comparing the group of subjects with SCA versus the group of subjects without.

For this purpose:

  1. We better clarify the combination of SCA and IMT as outcomes to stratify SCA stages (see introduction section (page 2, lines 51-59):

“…the preventive perspectives are complicated yet at early stages of atherosclerosis because of different factors. Firstly, presence of focal atherosclerotic lesions, detected by commonly used techniques in clinics (like ultrasound) identify subjects at increased risk of ACVD [1], although tracking Subclinical Carotid Atherosclerosis (SCA) progression by preclinical markers (including carotid Intima-Media Thickness (IMT)) remains a challenge to further stratify subjects in primary prevention [2]. Secondly, IMT and the presence of focal carotid vascular lesions (more robust indicator of ACVD risk [1]) are at the same time predicted by hardwired host-derived classical cardiovascular risk factors (CVRFs) (e.g. Type 2 Diabetes (T2D), Metabolic Syndrome (MetS), dyslipidemia), and patterns of individual predisposition to low-grade inflammation [3,4].…”.

References:

  1. Mach, F.; Baigent, C.; Catapano, A.L.; Koskinas, K.C.; Casula, M.; Badimon, L.; Chapman, M.J.; De Backer, G.G.; Delgado, V.; Ference, B.A.; et al. 2019 ESC/EAS Guidelines for the management of dyslipidaemias: Lipid modification to reduce cardiovascular risk. Eur. Heart J. 2020, 41, 111–188.

  1. Lorenz, M.W.; Polak, J.F.; Kavousi, M.; Mathiesen, E.B.; Völzke, H.; Tuomainen, T.-P.; Sander, D.; Plichart, M.; Catapano, A.L.; Robertson, C.M.; et al. Carotid intima-media thickness progression to predict cardiovascular events in the general population (the PROG-IMT collaborative project): a meta-analysis of individual participant data. Lancet (London, England) 2012, 379, 2053–62, doi:10.1016/S0140-6736(12)60441-3.

  1. Hoogeveen, R.M.; Pereira, J.P.B.; Nurmohamed, N.S.; Zampoleri, V.; Bom, M.J.; Baragetti, A.; Boekholdt, S.M.; Knaapen, P.; Khaw, K.-T.; Wareham, N.J.; et al. Improved cardiovascular risk prediction using targeted plasma proteomics in primary prevention. Eur. Heart J. 2020, doi:10.1093/eurheartj/ehaa648.

  1. Olmastroni, E.; Baragetti, A.; Casula, M.; Grigore, L.; Pellegatta, F.; Pirillo, A.; Tragni, E.; Catapano, A.L. Multilevel Models to Estimate Carotid Intima-Media Thickness Curves for Individual Cardiovascular Risk Evaluation. Stroke 2019, 50, 1758–1765, doi:10.1161/STROKEAHA.118.024692.

  1. We now better clarify definition of SCA in methods section (page 4, lines 134-163) as follows:

“…SCA was defined by ultrasound-based analysis of bilateral carotid arteries as previously described [27]. In detail, common carotid IMT (one centimeter from the bulb) was measured in longitudinal view, far wall, by high resolution B-mode ultrasound based system (Vivid S5 (GE Healthcareâ, Wauwatosa, WI, USA) connected to linear probe (4.0X13.0 MHz frequency; 14x48 mm footprint, 38 mm field of view)). A mean value for both sides was averaged. “+IMT” was distinguished by “-IMT” in presence of IMT above the 75th percentile of the median IMT for a Caucasian population according to ASE guidelines [29]. SCA was defined when mean IMT was ³ 1.3 mm or in presence of focal atherosclerotic lesions larger than 1.3 mm using a manual caliper in longitudinal view either in far or near wall and over every carotid tract (common, bulb section, bifurcation, internal or external branches). In two scans performed by the same operator in 75 subjects, the mean difference in IMT was 0.005±0.002 mm and the coefficient of variation (CV) was 1.93%. The correlation between two scans was significant (r = 0.96; P < 0.0001). The combination of information from IMT measurement and from presence/absence of SCA allowed to identified four different SCA stages: subjects without intimal thickening and without SCA (“-IMT/-SCA”, n=23); subjects with intimal thickening but without SCA (“+IMT/-SCA”, n=173); subjects without intimal thickening but with SCA (“-IMT/+SCA”, n=121); subjects with both intimal thickening and SCA (“+IMT/+SCA”, n=23).…

For Supplemental Figure 5 (E. coli metagenetic markers associated with advanced SCA progression), further vascular characterization according to validated criteria [30] allowed to distinguish, among the +IMT/+SCA group, subjects with “no advanced SCA” (stenosis <30% and P/S<125 cm/s) vs the “advanced SCA” (stenosis 30% and elevation of the P/S wave in the bilateral internal carotid branches). The advanced SCA was then divided by further characterization identifying: a) SCA causing stenosis between 30 and 50% with P/S <125 cm/s; b) SCA causing stenosis between 50 and 70% and P/S between 125 and 250 cm/s; c) SCA causing stenosis over 70% and P/S over 250 cm/s. We evaluated echolucencies of the atherosclerotic lesions among all subjects from the –IMT/+SCA and from the +IMT/+SCA group, using grey-scale definition and parameters of the QuickScanâ and autoIMTâ software included in the ultrasound machinery (Samsung HM70aâ, Samsungâ, Seoul, South Korea).”.

References:

  1. Baragetti, A.; Pisano, G.; Bertelli, C.; Garlaschelli, K.; Grigore, L.; Fracanzani, A.L.; Fargion, S.; Norata, G.D.; Catapano, A.L. Subclinical atherosclerosis is associated with Epicardial Fat Thickness and hepatic steatosis in the general population. Nutr. Metab. Cardiovasc. Dis. 2016, 26, doi:10.1016/j.numecd.2015.10.013.

  1. Stein, J.H.; Korcarz, C.E.; Hurst, R.T.; Lonn, E.; Kendall, C.B.; Mohler, E.R.; Najjar, S.S.; Rembold, C.M.; Post, W.S.; American Society of Echocardiography Carotid Intima-Media Thickness Task Force Use of Carotid Ultrasound to Identify Subclinical Vascular Disease and Evaluate Cardiovascular Disease Risk: A Consensus Statement from the American Society of Echocardiography Carotid Intima-Media Thickness Task Force Endorsed by the Society for Vascular Medicine. J. Am. Soc. Echocardiogr. 2008, 21, 93–111, doi:10.1016/j.echo.2007.11.011.
  2. Grant, E.G.; Benson, C.B.; Moneta, G.L.; Alexandrov, A. V.; Baker, J.D.; Bluth, E.I.; Carroll, B.A.; Eliasziw, M.; Gocke, J.; Hertzberg, B.S.; et al. Carotid Artery Stenosis: Gray-Scale and Doppler US Diagnosis - Society of Radiologists in Ultrasound Consensus Conference. In Proceedings of the Radiology; 2003; Vol. 229, pp. 340–346.

2) From the data included in Table 1, it appears that there are differences in subjects with or without SCA in several parameters, including Hepatic steatosis, uricemia, serum creatine, alcohol consumption, physical activity, lipid lowering and antihypertensive treatments. Would one not expect that changes in such parameters may also underlie some of the observed functional and taxonomic GM alterations?

We thank the reviewer for raising this further important aspect.

It should be noted that significant differences we observed in microbial composition comparing beta-diversity measures via Principal Coordinate Analysis (PCoA) still take into account potential confounders, since they are based on multivariate models, associated with data-reduction techniques that produce a set of uncorrelated (orthogonal) axes to summarize the variability in the data set.

This aspect has been now acknowledged in the limitations sections (page 15, lines 532-537):

“… The cross-sectional design of this investigation cannot rule out the burden of CVRFs prior to the examination, although significant differences we observed in microbial composition comparing beta-diversity measures via Principal Coordinate Analysis (PCoA) still take into account potential confounders, since they are based on multivariate models, associated with data-reduction techniques that produce a set of uncorrelated (orthogonal) axes to summarize the variability in the data set..”.

Nevertheless, according to Reviewer’ indication we now evaluated separately how much these parameters might interact in the relation between GM changes and SCA.

Pharmacological therapies (anti-hypertensive and lipid lowering drugs):

Firstly, the updated Table 1 actually reports that subjects with SCA were more on anti-hypertensive drugs. Of note, we did not found significant differences in either alpha (p>0.7 for all considered metrics) or beta diversities between subjects on anti-hypertensive as compared to those without anti-hypertensive (p=0.531 and p=0.479 for unweighted and weighted Unifrac distances, respectively). See attachment for representative plots for alpha- (left) and beta-diversity (right) plots.

We performed similar analysis for lipid lowering drugs (which were more prescribed in subjects with SCA, explaining their lower cholesterol and triglycerides levels as compared to those without SCA). Similarly, we did not find differences in either alpha (p=1 on all metrics) or beta diversities (p= 0.51 and p= 0.186 for unweighted and weighted Unifrac distances, respectively) between subjects on lipid lowering drugs as compared to those without lipid lowering drugs. See attachment for representative plots for alpha- (left) and beta-diversity (right) plots.

Based on these findings, we implemented this point in discussion section (page 14, lines 580-590), as follows:

“..Also the effect of pharmacological interventions over time (especially considering anti-hypertensive and lipid lowering drugs) cannot be excluded; in fact, we did not find differences in the GM alpha and beta diversities comparing subjects with pharmacological therapy and those without any treatments but we do not have precise information nor about their effect on GM composition neither on the bioavailability of these drugs (an aspect that could be particularly of interest for statins [23,63])…”.

References:

  1. Jie, Z.; Xia, H.; Zhong, S.-L.; Feng, Q.; Li, S.; Liang, S.; Zhong, H.; Liu, Z.; Gao, Y.; Zhao, H.; et al. The gut microbiome in atherosclerotic cardiovascular disease. Nat. Commun. 2017, 8, 845, doi:10.1038/s41467-017-00900-1.

  1. Tuteja, S.; Ferguson, J.F. Gut Microbiome and Response to Cardiovascular Drugs. Circ. Genomic Precis. Med. 2019, 12, 421–429.

Hepatic steatosis:

Hepatic steatosis was also more prevalent in subjects with SCA (who presented with increased waist-hip ratio versus those without SCA (p<0.001, see actual Table 1). We did not find differences in either alpha (p>0.578 on all metrics) or beta diversities (p=1 for both unweighted and weighted Unifrac distances, respectively). See attachment for representative plots for alpha- (left) and beta-diversity (right) plots.

Of note, when repeating the analyses including only subjects without hepatic steatosis, we further confirmed differences in alpha and beta diversities between subjects with SCA versus those without SCA, but it is to be acknowledged that the poor number of subjects with hepatic steatosis (n=31) might underrepresents the possible effect of this parameter on GM composition.

Based on this finding, we implemented this point in discussion section (page 14, lines 521-523), as follows:

“…Also, hepatic steatosis (which was only ultrasound-determined and poorly prevalent in our cohort) as well as plasma hs-CRP, produced by hepatocytes marking low-grade inflammation, did not correlate either with GM diversity (while recently reported in morbidly obese patients [46] and in liver steatosis [47]), or with reduced relative abundance of specific bacterial species (e.g.: Akkermansia muciniphila (less abundant in insulin resistance [48] and in obese individuals [49]). Although these data might exclude an impact of a gut-liver connection during early stages of SCA, in depth evaluation on larger cohorts with more advanced stages of liver disease should are warranted..”.

References:

  1. Vieira-Silva, S.; Falony, G.; Belda, E.; Nielsen, T.; Aron-Wisnewsky, J.; Chakaroun, R.; Forslund, S.K.; Assmann, K.; Valles-Colomer, M.; Nguyen, T.T.D.; et al. Statin therapy is associated with lower prevalence of gut microbiota dysbiosis. Nature 2020, 1–6, doi:10.1038/s41586-020-2269-x.

  1. Aron-Wisnewsky, J.; Gaborit, B.; Dutour, A.; Clement, K. Gut microbiota and non-alcoholic fatty liver disease: New insights. Clin. Microbiol. Infect. 2013, 19, 338–348.

  1. Schneeberger, M.; Everard, A.; Gómez-Valadés, A.G.; Matamoros, S.; Ramírez, S.; Delzenne, N.M.; Gomis, R.; Claret, M.; Cani, P.D. Akkermansia muciniphila inversely correlates with the onset of inflammation, altered adipose tissue metabolism and metabolic disorders during obesity in mice. Sci. Rep. 2015, 5, doi:10.1038/srep16643.

  1. Dao, M.C.; Everard, A.; Aron-Wisnewsky, J.; Sokolovska, N.; Prifti, E.; Verger, E.O.; Kayser, B.D.; Levenez, F.; Chilloux, J.; Hoyles, L.; et al. Akkermansia muciniphila and improved metabolic health during a dietary intervention in obesity: Relationship with gut microbiome richness and ecology. Gut 2016, 65, 426–436, doi:10.1136/gutjnl-2014-308778.

Finally, we also did not find significant interaction between GM diversities with uric acid (increased in SCA) and prevalence of self-reported physical activity (reduced in SCA). Since uric acid is tightly related with the other parameters previously analyzed and since the prevalence of physical activity was self-reported by subjects but not validated by quantitative tools, we believe that future and better focused analyses on these aspects are required to draw more robust conclusions.

3) In Table 1, please clarify which comparisons were made for statistical analysis (p values).

Table 1 now indicates numerical P value referring to comparison between subjects with SCA and those without SCA. Moreover, asterisks highlight significant differences (“***” indicates P<0.005; “**” indicates P<0.01; “*” indicates P<0.05).

4) Use of the term SCA progression may be slightly misleading as the authors are not analyzing samples of the same patient at different time points. Maybe the used of another term, eg. SCA stages, could be more appropriate.

We thank the reviewer for this point and we agree that the cross-sectional nature of this investigation cannot imply a progression of SCA. We thus changed the term “SCA progression” into “SCA stages” over the text accordingly.

5) Based on the coloring scheme used in the heat map of Fig4, there doesn’t seem to be any major change in dietary habits between no SCA and SCA groups. This is in contrast to the description of the authors finding regarding significant changes in dietary habits (page 11, lines 289-295). Also, please revise the figure so that the label ‘avg_No SCA’ is shown a single line as in its current state this labeling is misleading by implying that there are 2 samples.

In accordance to reviewer’ suggestion, we now converted the heatmap as barplot showing significantly different intakes of dietary patterns between subjects with SCA vs those without SCA subjects (Figure 4A).

In addition, we now propose an updated version of the biplot in figure 4C, which highlights thirteen dietary patterns co-segregating with GM taxonomic differences between subjects with SCA versus those without SCA.

6) According to information on Supplementary Table 4, the SCA group shows significantly increased consumption of ‘’mechanically separated meats’’, but no statistical difference in the consumption of ‘’Meat, meat products and substitutes’’ or ‘’Beef, veal, poultry and pork, not preserved’’. This is in contrast to what is mentioned in lines 289-290 of page 11 ‘’Subjects with SCA reported to consume higher amounts of mechanically separated meats (p=0.016) and beef, veal, poultry and pork’’. Please correct text accordingly.

We apologize for previous misreporting and we now amended this point in the result section (which has been now re-scheduled as 3.3 “Individual diet clusters with changes in taxonomic GM composition and SCA”) (page 12, lines 400-407) as follows:

“…Subjects with SCA reported to consume higher amounts of mechanically separated meats (p=0.013) although a similar amount of ham, salami, sausages (p=0.414), meat products and substitutes (p=0.470) and a similar amount of not preserved meat (beef, veal, poultry and pork) (p=0.683) as compared to subjects without SCA. Moreover, they reported to consume more dried fruits (p=0.02), as well as a trend towards higher quantities of fruits (both processed and fresh) and eggs. By contrast, subjects without SCA reported to consume a higher amount of cereals (p=0.009), starchy vegetables (p=0.027), milky products and beverages (p=0.004 and p=0.016, respectively), yoghurts (p=0.047) and bakery (p<0.001) as compared to those with SCA (Figure 4A).”.

Moreover, we revised this aspect in discussion section (page 15, lines 503-512), as follows:

“…Vice versa, we found a more over-expressed biosynthetic pathway of PE (strictly linked to the hepatic conversion to atherogenic TMAO [15] mediated by the L-carnitine/γBB/TMA metabolic cascade [56]), together with increased monocytes and neutrophils counts in subjects with SCA, who reported to consume only higher amount of mechanically separated meats but a similar amount of unprocessed meats (as more complex food matrices containing other nutrients, phospholipids and probiotics that have been not associated with higher ACVD risk [57]) versus subjects without SCA. Future analyses and interventional dietary approaches will be necessary to unveil whether these connections are more likely reflecting a gut-bone marrow connection is set fostering persistent patrolling activity of the innate immunity against dietary fats and bacterial lipid products.…”.

References:

  1. Tang, W.H.W.; Wang, Z.; Fan, Y.; Levison, B.; Hazen, J.E.; Donahue, L.M.; Wu, Y.; Hazen, S.L. Prognostic value of elevated levels of intestinal microbe-generated metabolite trimethylamine-N-oxide in patients with heart failure: Refining the gut hypothesis. J. Am. Coll. Cardiol. 2014, 64, 1908–1914, doi:10.1016/j.jacc.2014.02.617.
  2. van der Veen, J.N.; Kennelly, J.P.; Wan, S.; Vance, J.E.; Vance, D.E.; Jacobs, R.L. The critical role of phosphatidylcholine and phosphatidylethanolamine metabolism in health and disease. Biochim. Biophys. Acta - Biomembr. 2017, 1859, 1558–1572.
  3. Astrup, A.; Magkos, F.; Bier, D.M.; Brenna, J.T.; de Oliveira Otto, M.C.; Hill, J.O.; King, J.C.; Mente, A.; Ordovas, J.M.; Volek, J.S.; et al. Saturated Fats and Health: A Reassessment and Proposal for Food-Based Recommendations: JACC State-of-the-Art Review. J. Am. Coll. Cardiol. 2020, 76, 844–857.

7) In Supplemental Figure 4 there is no data on increased plasma levels of zonulin in +IMT/+SCA subjects, as mentioned in the Discussion page 14, line 407. Instead this figure shows data on E. coli metagenetic markers associated with advanced SCA progression. Please correct.

We apologize for missing information in the previous form and we now implemented data on increased levels of zonulin as Supplemental Figure 6 (“Figure S6: Plasma zonulin levels are not increased in subjects with +IMT/+SCA.).

8) In Figure 5, the MetaCyc L4-pathways and the bacterial species of the two heat-maps are too small to read. Given that this is important information, the authors should consider increasing the font or showing only some of the most significant changes. Findings shown in this figure are important but currently they are not visible.

We now improved graphical presentation of the figure (which has been now re-scheduled as Figure 3), by rotating to 90° clockwise and increasing font size.

The figure now should be printed full page and should be more readable.

9) Do the authors have any other experimental evidence on bacterial metabolites (eg from serum) to confirm the observed alterations in metagenomic pathways?

We completely agree with the reviewer that analysis of circulating metabolomics markers from patients would have been an important additional information on top of metagenomic analyses.

We previously found, by machine learning-based data from the same PLIC cohort, a specific innate immune-inflammatory proteomic set significantly predicting faster progression of SCA over time (see Ref#3 in the text).

Since we found some immune-inflammatory markers increased in subjects with SCA (monocytes) and +IMT/+SCA (neutrophils), actually a more comprehensive untargeted proteomics and metabolomics crossing with metagenomic data would be of high scientific value for this purpose. This will be considered as a future perspective for further analyses on this same cohort.

We discussed this point in discussion section (page 14, lines 437-441), as follows:

“…. Finally, we did not perform a complete analysis of circulating proteomic and metabolomics markers validating the metagenomics pathways that emerged from our data. Since multiple circulating proteins and metabolites (the majority of which of lipid origin) have been recently associated with different types of dietary patterns [64], this aspect is of relevance and will be analyzed in the near future.…”.

Reference:

  1. Walker, M.E.; Song, R.J.; Xu, X.; Gerszten, R.E.; Ngo, D.; Clish, C.B.; Corlin, L.; Ma, J.; Xanthakis, V.; Jacques, P.F.; et al. Proteomic and metabolomic correlates of healthy dietary patterns: The framingham heart study. Nutrients 2020, 12, doi:10.3390/nu12051476.

10) According to this study, taxonomic changes in GM composition were shown to occur during first stages of subclinical atherosclerosis. The authors conclude that this coincides with individual exposure to dietary sources, highlighting functional metagenomic relevance. However, it is still unclear to me how GM composition is correlated with dietary patterns. Please explain in section 3.2 the analysis performed for this and describe in more detail the findings. This is an important point of this manuscript but it is not adequately explained. A figure for this should also be included.

We thank the reviewer for raising this seminal aspect of the manuscript that, we agree, it was not clearly described in the previous form.

We now provided additional analyses, correlating dietary patterns with GM components in subjects without SCA and in those with SCA (see Figure 4B and attachment). Of note we found a higher number of significant correlations in presence of SCA and with relative abundances of multiple genera that emerged significant from the metagenomic analysis (see Faecalibacterium, and Escherichia).

Finally, we found up to thirteen specific food patterns significantly co-segregating with the GM beta diversities characterizing subjects with SCA versus those without SCA (see Figure 4C and attachment).

Although this cross-sectional analysis cannot draw conclusions about the relation of causality, we believe that these data can be considered as preliminary indicators of a relation between different GM re-shape in response to different exposure to dietary patterns over SCA stages.

This aspect has been revised:

  1. re-scheduling the order of the paragraphs in results section; we switched paragraph “Individual diet clusters with changes in taxonomic GM composition and SCA” (now as 3.3) with the paragraph “Functional relevance of GM dysbiosis over subclinical carotid atherosclerosis” (now as 3.2).
  2. re-phrasing 3.3 paragraph (page 12, lines 400-423) as follows:

“..Subjects with SCA reported to consume higher amounts of mechanically separated meats (p=0.013) although a similar amount of ham, salami, sausages (p=0.414), meat products and substitutes (p=0.470) and a similar amount of not preserved meat (beef, veal, poultry and pork) (p=0.683) as compared to subjects without SCA. Moreover, they reported to consume more dried fruits (p=0.02), as well as a trend towards higher quantities of fruits (both processed and fresh) and eggs. By contrast, subjects without SCA reported to consume a higher amount of cereals (p=0.009), starchy vegetables (p=0.027), milky products and beverages (p=0.004 and p=0.016, respectively), yoghurts (p=0.047) and bakery (p<0.001) as compared to those with SCA (Figure 4A).

Moreover, these dietary intakes showed more significant correlations to bacterial genera in subjects with SCA as compared to those without SCA (Figure 4B; red square represent positive correlation while blue square indicates negative correlation; yellow dots indicate significant correlations, p<0.05). Of note, some of these correlation recapitulated data from metagenomic analysis. For example, increased relative abundance of Escherichia was inversely related with intakes of yoghurts, pulses and fresh vegetables while positively correlated to bakery only in individuals with SCA.

It is of note that we were overall able to cluster intakes of particular food patterns (including milky products, yogurts, total and leafy vegetables, processed cereals, mechanically separated meats, fish, bread and bakery products) that significantly co-segregated with the taxonomic diversities between subjects with SCA as compared to subjects without SCA (RV coefficient between nutrients quantities and microbial relative abundances at genus level=0.65, p-value=0.047) (Figure 4C). In addition to this finding, significant changes (p=0.04) in the individual diet were more significantly evident in subjects with +IMT/+SCA versus those without subclinical atherosclerosis (-IMT/-SCA) (not shown), further supporting interaction of dietary exposure on GM changes and early stages of SCA..”.

11) Along these lines, would it not have been more appropriate if all subjects had been under similar dietary habits? Would this not represent a more homogeneous and potentially better strategy to study the functional role of the different GM composition in SCA progression?

The purpose of this study was to primarily investigate if GM taxonomic and metagenomic changes occur during initial stages of SCA in an epidemiological study, gathering data of self-reported dietary habits. Although this approach might represent a good model of real-life setting, we agree with the reviewer that clustering a priori the population by different dietary habits would be an alternative strategy. However, longitudinal evaluations are needed to unveil the actual effect of a single dietary pattern/habit on GM composition/functionality in subjects with different SCA stages.

We thank the reviewer for this suggestion which prompt us to design tailored analyses in the near future and we included this aspect in the discussion section (page 15, lines 523-529):

“…Firstly, this epidemiological study, gathering self-reported dietary data, does not allow to unveil the actual relation of causality. These limitations pave the road to dissect this aspect in the near future, a perspective that might be pursued: i) by clustering larger number of subjects one the basis of their exposure to different dietary patterns (daily collected using smartphone health apps or other health mobile technologies to improve self-reporting) or ii) by longitudinally evaluating the actual effect of a single dietary pattern/habit on changes of GM composition/functionality in subjects with different SCA stages. …”.

12) Given that the study includes individuals of subclinical stages of ACVD, what do the authors believe is the most significant change observed at this stage of disease in relation to GM composition, particularly dietary pattern, or specific metagenomic pathway alteration? Do the authors envision that reversal of such change may result in prevention of disease progression?

We thank again the reviewer for this important point.

In line with previous comment, the cross-sectional design of this investigation cannot rule out the direction of causality between dietary pattern and GM taxonomic or functional composition. Vice versa this aspect would be better addressed through a cross-over trial analyzing the effect of dietary intervention on GM composition and on markers of SCA progression over time.

This appears an appealing task in the cardiovascular area, but the design and the multiple technical hurdles of lifestyle and dietary intervention studies make difficult the interpretation of their resulting data and their export to wider populations (see Ref#24).

As per reviewer’ suggestion, we now have implemented this limitation in discussion section (page 15, lines 530-532):

“…Different hurdles undermining these approaches are to be accounted in the design of lifestyle and dietary intervention studies (e.g.: technical criticisms, difficulties in the interpretation of their resulting data and the export to wider populations [24]).…”.

Reference:

  1. Swann, J.R.; Rajilic-Stojanovic, M.; Salonen, A.; Sakwinska, O.; Gill, C.; Meynier, A.; Fança-Berthon, P.; Schelkle, B.; Segata, N.; Shortt, C.; et al. Considerations for the design and conduct of human gut microbiota intervention studies relating to foods. Eur. J. Nutr. 2020, 59, 3347–3368.

13) Since there is a lot of bionformatical analysis and graphs included in the article, please ensure to clearly explain what samples were analyzed in each graph, why this was done and what is the main findings from each analysis. For example, there are several PCA plots included but it is not always clear as to what each one of them is showing.

Following the reviewer’s suggestion, a critical review of the text and data presentation has been performed, trying to make clear and point-by-point which figure/table support each described result. Moreover, figures legends were also re-phrased including additional information about data analysis.

14) There appears to be a lot of information (methods and results) as supplementary. This makes it very disruptive to the reader, by constantly having to refer to supplementary material, but also difficult to follow due to the large amount of information included. It may be best to condense the supplementary material and include some of the most important information in the main text.

We agree with the reviewer and some information from supplemental Material has been moved to the main text.

In detail:

  1. We imported in method sections, the exclusion criteria and data collection of clinical and biochemical information, at pages 3-4, lines 110-132:

“.. Additionally, we excluded: i) subjects reporting use of glucose-lowering drugs, (ii) with positive personal history of CVD (either ischemic heart disease, ST elevation or non-ST elevation myocardial infarction, aortic-coronary by-pass grafting, angioplasty, transient ischemic attack, stroke, heart failure from Class II to IV according to New York Heart Association (NYHA) definition or documented peripheral arteriopathy), (iii) with MetS (defined according to harmonized criteria of the American Heart Association47), iv) chronic kidney disease (GFR < 60 ml/min or documented albuminuria > 30 mg/g), iv) pregnancy and v)  reported malignancies. Data management and statistical analyses were performed with the coordination of the Epidemiology and Preventive Pharmacology Centre (SEFAP) of the University of Milan. The study was approved by the Scientific Committee of the University of Milan (SEFAP/Pr.0003). An informed consent was obtained by subjects (all over 18 years-old), in accordance with the Declaration of Helsinki. Systolic and diastolic blood pressure and Body Mass Index (BMI), waist and waist/hip ratio were measured. Information on presence of hepatic steatosis, available on a subgroup of 133 subjects, was defined via ultrasound, as per already published protocols. Blood samples were collected from antecubital vein after 12 hours fasting on NaEDTA tubes (BD Vacuette) and, then, centrifuged at 3,000 rpm for 12 minutes (Eppendorf 580r, Eppendorf, Hamburg, Germany) for biochemical parameters profiling including: total cholesterol, HDL-C, triglycerides, ApoB, ApoA-I, glucose, liver enzymes, creatinine and creatinine-phospho kinase (CPK). Measurements were performed using immuno-turbidimetric and enzymatic methods thorough automatic analyzers (Randox, Crumlin, UK). LDL-C was derived from Friedewald formula. Separately, whole blood in NaEDTA tubes was used for haematocrit analysis to derive total count of leukocytes and their fractions (neutrophils, lymphocytes, monocytes, eosinophils and basophils, indicated as cells*1000/microliter).”.

References:

  1. Fracanzani, A.L.; Pisano, G.; Consonni, D.; Tiraboschi, S.; Baragetti, A.; Bertelli, C.; Norata, G.D.; Dongiovanni, P.; Valenti, L.; Grigore, L.; et al. Epicardial Adipose Tissue (EAT) thickness is associated with cardiovascular and liver damage in nonalcoholic fatty liver disease. PLoS One 2016, 11, doi:10.1371/journal.pone.0162473.

  1. Alberti, K.G.M.M.; Eckel, R.H.; Grundy, S.M.; Zimmet, P.Z.; Cleeman, J.I.; Donato, K.A.; Fruchart, J.C.; James, W.P.T.; Loria, C.M.; Smith, S.C. Harmonizing the metabolic syndrome: A joint interim statement of the international diabetes federation task force on epidemiology and prevention; National heart, lung, and blood institute; American heart association; World heart federation; International atherosclerosis society; And international association for the study of obesity. Circulation 2009, 120, 1640–1645.

  1. We imported Lifestyle data, collection and analysis of dietary habits at page 4-5, lines 167-186:

“.. Subjects self-reported their level and type of physical activity and smoking habit and information about individual diet were collected in the PLIC Study as previously reported [31]. In detail, all subjects were requested to complete a semi-quantitative daily food diary representative of seven days before the clinical evaluation and collection of fecal sample. The food diary was administered to subjects following instructions about the reporting of quali/quantitative dietary information by two dieticians (blinded on subject’s clinical history). In the food diary, subjects reported for each meal (breakfast, lunch, dinner and snacks) the foods, the brand names of foods (where applicable), the methods of preparation and dressings. During the seven days, dieticians were available for help and to provide more instructions to subjects by phone or by email. A portion reference from validated color photographs (the “Atlante Fotografico delle Porzioni degli Alimenti”; https://www.scottibassani.it/atlante-fotografico-delle-porzioni-degli-alimenti/) was also given to subjects, for further help in the interpretation of food quantities. Then, after seven days, the filled out food diary was analyzed by dieticians during the outpatient evaluation in front of the subject: i) to clarify details and improper indications and ii) to derived individual daily energy and the seven-day dietary records averaged nutrient intakes (as g/week), referring to the Italian BDA database (www.bda-ieo.it. BDA- Food Composition Database for Epidemiological Studies in Italy (2015)). Also, BDA and the reference values of the Italian Society of Nutrition (“LARN”, Livelli di Assunzione di Riferimento di Nutrienti ed Energia) were used to exclude outlier data about energy intake, deriving from improper self-reporting of the subject.

Reference:

  1. Redaelli, L.; Garlaschelli, K.; Grigore, L.; Norata, G.D.; Catapano, A.L. Association between the Adherence to AHA Step 1 Nutrition Criteria and the Cardiometabolic Outcome in the General Population a Two Year Follow-Up Study. Food Nutr. Sci. 2012, 3.

  1. We imported information about ultrasound-based vascular characterization at pages 3-4, lines 141-163:

“…SCA was defined by ultrasound-based analysis of bilateral carotid arteries as previously described [27]. In detail, common carotid IMT (one centimeter from the bulb) was measured in longitudinal view, far wall, by high resolution B-mode ultrasound based system (Vivid S5 (GE Healthcareâ, Wauwatosa, WI, USA) connected to linear probe (4.0X13.0 MHz frequency; 14x48 mm footprint, 38 mm field of view)). A mean value for both sides was averaged. “+IMT” was distinguished by “-IMT” in presence of IMT above the 75th percentile of the median IMT for a Caucasian population according to ASE guidelines [29].

SCA was defined when mean IMT was ³1.3 mm or in presence of focal atherosclerotic lesions larger than 1.3 mm using a manual caliper in longitudinal view either in far or near wall and over every carotid tract (common, bulb section, bifurcation, internal or external branches). In two scans performed by the same operator in 75 subjects, the mean difference in IMT was 0.005±0.002 mm and the coefficient of variation (CV) was 1.93%. The correlation between two scans was significant (r = 0.96; P < 0.0001). The combination of information from IMT measurement and from presence/absence of SCA allowed to identified four different SCA stages: subjects without intimal thickening and without SCA (“-IMT/-SCA”, n=23); subjects with intimal thickening but without SCA (“+IMT/-SCA”, n=173); subjects without intimal thickening but with SCA (“-IMT/+SCA”, n=121); subjects with both intimal thickening and SCA (“+IMT/+SCA”, n=23).

Whole shotgun metagenomic sequencing analyses were performed on the same fecal samples of 23 “-IMT/-SCA” and 23 “+IMT/+SCA”, whose clinical characteristics are reported in (Supplemental Table 1). For Supplemental Figure 5 (E. coli metagenetic markers associated with advanced SCA progression), further vascular characterization according to validated criteria [30] allowed to distinguish, among the +IMT/+SCA group, subjects with “no advanced SCA” (stenosis <30% and P/S<125 cm/s) vs the “advanced SCA” (stenosis 30% and elevation of the P/S wave in the bilateral internal carotid branches). The advanced SCA was then divided by further characterization identifying: a) SCA causing stenosis between 30 and 50% with P/S <125 cm/s; b) SCA causing stenosis between 50 and 70% and P/S between 125 and 250 cm/s; c) SCA causing stenosis over 70% and P/S over 250 cm/s. We evaluated echolucencies of the atherosclerotic lesions among all subjects from the –IMT/+SCA and from the +IMT/+SCA group, using grey-scale definition and parameters of the QuickScanâ and autoIMTâ software included in the ultrasound machinery (Samsung HM70a, Samsungâ, Seoul, South Korea).”.

References:

  1. Baragetti, A.; Pisano, G.; Bertelli, C.; Garlaschelli, K.; Grigore, L.; Fracanzani, A.L.; Fargion, S.; Norata, G.D.; Catapano, A.L. Subclinical atherosclerosis is associated with Epicardial Fat Thickness and hepatic steatosis in the general population. Nutr. Metab. Cardiovasc. Dis. 2016, 26, doi:10.1016/j.numecd.2015.10.013.

  1. Stein, J.H.; Korcarz, C.E.; Hurst, R.T.; Lonn, E.; Kendall, C.B.; Mohler, E.R.; Najjar, S.S.; Rembold, C.M.; Post, W.S.; American Society of Echocardiography Carotid Intima-Media Thickness Task Force Use of Carotid Ultrasound to Identify Subclinical Vascular Disease and Evaluate Cardiovascular Disease Risk: A Consensus Statement from the American Society of Echocardiography Carotid Intima-Media Thickness Task Force Endorsed by the Society for Vascular Medicine. J. Am. Soc. Echocardiogr. 2008, 21, 93–111, doi:10.1016/j.echo.2007.11.011.

  1. Grant, E.G.; Benson, C.B.; Moneta, G.L.; Alexandrov, A. V.; Baker, J.D.; Bluth, E.I.; Carroll, B.A.; Eliasziw, M.; Gocke, J.; Hertzberg, B.S.; et al. Carotid Artery Stenosis: Gray-Scale and Doppler US Diagnosis - Society of Radiologists in Ultrasound Consensus Conference. In Proceedings of the Radiology; 2003; Vol. 229, pp. 340–346.

  1. We imported dietary data analysis (page 6, lines 244-249):

“..Statistical data of nutrients composition for individuals with and without SCA was performed by employing the non-parametric Mann-Whitney U-test. Overall separation between patients was assessed calculating Bray-Curtis distances among patients on the basis of the nutrients table and “adonis” function in the R package “vegan” was used. In order to assess the correlation between dietary and microbial composition data, the RV coefficient [38] was calculated; coefficient statistical significance was calculated by 99999 random permutations [39]”.

References:

  1. Smilde, A.K.; Kiers, H.A.L.; Bijlsma, S.; Rubingh, C.M.; Van Erk, M.J. Matrix correlations for high-dimensional data: The modified RV-coefficient. Bioinformatics 2009, 25, 401–405, doi:10.1093/bioinformatics/btn634.

  1. Josse, J.; Pagès, J.; Husson, F. Testing the significance of the RV coefficient. Comput. Stat. Data Anal. 2008, 53, 82–91, doi:10.1016/j.csda.2008.06.012.

  1. Parts of the “Microbiome 16s data analysis” are now described in more detail in supplemental material:

“..The 16S rRNA raw sequences were processed through paired-end reads merging by PANDAseq55 and discarding low quality reads (i.e., showing stretches of bases with a Q-score <3 for more than 25% of their length). In order to obtain a similar number of reads for each sample, a subset of 50,000 reads per sample was randomly extracted. Bioinformatic analyses were conducted using the QIIME pipeline (release 1.8.056); filtered reads were clustered into Operational Taxonomic Units (OTUs) at 97% similarity level and taxonomically assigned via the RDP classifier 57 against the Greengenes database (release 13.8; (ftp://greengenes.microbio.me/greengenes_release/gg_13_8_otus), with a 0.5 identity threshold.

Biodiversity and distribution of the microorganisms were characterized via alpha- and beta-diversity evaluations. Alpha-diversity was measured using Chao1, observed species, Shannon diversity, Good's coverage and Faith's phylogenetic diversity (PD_whole_tree) metrics; statistical evaluation among alpha-diversity indices was performed by a non-parametric Monte Carlo-based test, using 999 random permutations.

In order to compare the microbial community structure in beta-diversity analysis, we calculated weighted and unweighted UniFrac distances 58 and performed principal coordinates analysis (PCoA); and statistical significance of the separation was assessed by analysis of variance with partitioning among sources of variation, using a permutation test with pseudo-F ratios (“adonis” function) in the R package “vegan” (version 2.0–10; https://cran.r-project.org/src/contrib/Archive/vegan/vegan_2.0-10.tar.gz).

Differences in abundances of bacterial taxa among experimental groups were analyzed by non-parametric Mann-Whitney U-test using MATLAB software (Natick, MA, USA). Unless otherwise stated, p < 0.05 were considered as significant for each statistical analysis.”.

  1. Parts of the “metagenome data analysis” are now described in more detail in supplemental material:

“In order to remove low quality and human sequence, shotgun metagenomic reads were quality filtered using the human sequence removal pipeline (https://www.hmpdacc.org/hmp/doc/HumanSequenceRemoval_SOP.pdf) and the processing procedures (https://www.hmpdacc.org/hmp/doc/ReadProcessing_SOP.pdf) from the Human Microbiome Project 59. Resulting reads were, then, processed by HUMAnN2 pipeline (v. 0.11.2, 60), which performs a nucleotide- and a translated-based search of reads against UniRef 90 protein database 61. Proteins were assigned to metabolic reactions and pathways via annotation on the MetaCyc pathways database of primary and secondary metabolism 62. In order to compensate for different sequencing depths, all measures were expressed as copies-per-million (CPM). Taxonomic profile of the samples were derived from the Metaphlan (v. 2.7.7, 63) alignment within HUMAnN2 pipeline.

Alpha-diversity evaluation was performed on species-level taxonomic classification and MetaCyc reaction-level functional classification, using non-phylogenetic indexes (i.e.: Chao1, observed_species and Simpson’s index) and a permutation-based Monte Carlo -based test with 999 random permutations in QIIME. Similarly, beta-diversity analysis was performed on Bray-Curtis and Euclidean distances and differences were assessed by “adonis” test with 999 random permutations in QIIME. Pathways were grouped to upper levels thanks to their lineage association in MetaCyc.”.

We believe that now all the principal findings and methods are in the main text of the paper and that Supplemental Material would be helpful for those really interested in some relatively less pivotal aspects.

Reviewer 2 Report

The submitted article presents a very relevant and current topic. The relationship of Gut Microbiota with several biochemical marks of cardiovascular disease has been studied, but the present article differs due to the fact that it study patients in a subclinical state.

In general, the article is very well written, however I have some suggestions for improvement in the Results and Discussion section:

It should be possible to read tables and figures without having to resort to the text of the article. Thus, all acronyms must be properly identified in the footer of the Table/Figure and the significance of the statistical tests must also be identified in the footer or caption.

In point 3.2, line 287 indicates that a food diary has been applied. How many hours (72h?) Should be described and whether the necessary instructions for completing it have been given. Was it a food diary with weighing food with a scale or estimated food intake by portions in homemade measures? Did all participants that complete the diary have valid results? Was there a follow-up during the week of the records to clarify doubts? That information must be given.

Figure 5 is too small and its correct reading is not possible. It must be bigger.

Lines 385 and 386: A comparison of the Clostridium species results is made with other studies in which patients with diabetes and obese were evaluated. However, in Table 1 we can see that the average BMI is 26.44, which means the presence of excess weight. This point should be reformulated with an indication of the prevalence of obesity and pre-obesity in the sample so that it can correctly compare with other studies carried out in obese individuals.

The authors identify some limitations of the study but I consider that an unidentified limitation is the fact that there are significant differences between the average age of the groups with and without carotid atherosclerosis. Because they are younger, they may not have it yet. There may be a confounding factor here. Clarification on this aspect should be added in the limitations of the study.

Author Response

Dear Reviewer#2,

Thank you very much for your positive and constructive comments. We have now revised the manuscript according to your suggestions and you can find below the responses to each of your comments.

The submitted article presents a very relevant and current topic. The relationship of Gut Microbiota with several biochemical marks of cardiovascular disease has been studied, but the present article differs due to the fact that it study patients in a subclinical state.

In general, the article is very well written, however I have some suggestions for improvement in the Results and Discussion section:

It should be possible to read tables and figures without having to resort to the text of the article. Thus, all acronyms must be properly identified in the footer of the Table/Figure and the significance of the statistical tests must also be identified in the footer or caption.

Tables in the main text have been now rescheduled in order to improve their quality for reading.

P values have been changed into symbols referring for details about statistical significance levels between groups in the caption.

In point 3.2, line 287 indicates that a food diary has been applied. How many hours (72h?) Should be described and whether the necessary instructions for completing it have been given. Was it a food diary with weighing food with a scale or estimated food intake by portions in homemade measures? Did all participants that complete the diary have valid results? Was there a follow-up during the week of the records to clarify doubts? That information must be given.

We agree with the reviewer that additional information about dietary intakes are required.

Individual dietary habits in PLIC Study were collected by the use of a semi-quantitative daily food diary representative of seven days that each subject filled out before the outpatient examination with dieticians (blinded on subject’ clinical history). The semi-quantiative daily food dairy reports portions in homemade measures. In order to minimize improper self-reporting:

  1. i) each subject in PLIC was provided with a portion reference from validated color photographs (the “Atlante Fotografico delle Porzioni degli Alimenti”; https://www.scottibassani.it/atlante-fotografico-delle-porzioni-degli-alimenti/)

  1. ii) dieticians analyzed the diary during outpatient evaluation using Italian database for food compositions (BDA: www.bda-ieo.it. BDA- Food Composition Database for Epidemiological Studies in Italy (2015)) and the reference values of the Italian Society of Nutrition (“LARN”, Livelli di Assunzione di Riferimento di Nutrienti ed Energia) to exclude outlier data about energy intaks.

Accordingly, we have now moved from supplemental material to the main text, implementing more methodological information about collection of dietary habits for all the subjects included in the study (page 4-5, lines 167-186:

“.. Subjects self-reported their level and type of physical activity and smoking habit and information about individual diet were collected in the PLIC Study as previously reported [31]. In detail, all subjects were requested to complete a semi-quantitative daily food diary representative of seven days before the clinical evaluation and collection of fecal sample. The food diary was administered to subjects following instructions about the reporting of quali/quantitative dietary information by two dieticians (blinded on subject’s clinical history). In the food diary, subjects reported for each meal (breakfast, lunch, dinner and snacks) the foods, the brand names of foods (where applicable), the methods of preparation and dressings. During the seven days, dieticians were available for help and to provide more instructions to subjects by phone or by email. A portion reference from validated color photographs (the “Atlante Fotografico delle Porzioni degli Alimenti”; https://www.scottibassani.it/atlante-fotografico-delle-porzioni-degli-alimenti/) was also given to subjects, for further help in the interpretation of food quantities. Then, after seven days, the filled out food diary was analyzed by dieticians during the outpatient evaluation in front of the subject: i) to clarify details and improper indications and ii) to derived individual daily energy and the seven-day dietary records averaged nutrient intakes (as g/week), referring to the Italian BDA database (www.bda-ieo.it. BDA- Food Composition Database for Epidemiological Studies in Italy (2015)). Also, BDA and the reference values of the Italian Society of Nutrition (“LARN”, Livelli di Assunzione di Riferimento di Nutrienti ed Energia) were used to exclude outlier data about energy intake, deriving from improper self-reporting of the subject.

Reference:

  1. Redaelli, L.; Garlaschelli, K.; Grigore, L.; Norata, G.D.; Catapano, A.L. Association between the Adherence to AHA Step 1 Nutrition Criteria and the Cardiometabolic Outcome in the General Population a Two Year Follow-Up Study. Food Nutr. Sci. 2012, 3.

Figure 5 is too small and its correct reading is not possible. It must be bigger.

Following the reviewer’s suggestion, the figure (now presented as Figure 3) has been rotated and the font size increased.

Lines 385 and 386: A comparison of the Clostridium species results is made with other studies in which patients with diabetes and obese were evaluated. However, in Table 1 we can see that the average BMI is 26.44, which means the presence of excess weight. This point should be reformulated with an indication of the prevalence of obesity and pre-obesity in the sample so that it can correctly compare with other studies carried out in obese individuals.

We now implemented this information in the actual Table 1, now showing a major prevalence of lean (BMI < 25 Kg/m2) and overweight subjects (25 < BMI < 30 Kg/m2) (34.4% and 49.4% respectively, out of total data available of the cohort) while a significantly minor prevalence of obesity (BMI > 30 Kg/m2) (15.8% out of total data available of the cohort).

(See Table in the attachment).

We did not find a correlation between BMI and the relative abundance of Clostridium in contrast to what found in three different studies, speculating that this could be explained by the different metabolic phenotype of our population, where T2D and MetS clinically defined were excluded.

According to reviewer’ suggestion we now clarify this aspect in discussion section (page 15, lines-411-418), as follows:

“…We analyzed the GM profile in 345 subjects (the majority of whom aged between 60 and 80 years-old), prevalently lean/overweight, without clinically manifest ACVD, T2D and MetS. Taxonomic compositions taken from this cohort (one of the largest of this kind) was comparable to those found in other geographically distant Italian cohorts [41,42] although did not confirm some correlations between relative abundances of genera/species and cardio-metabolic markers (Supplemental Figure 4, Supplemental Material), that were previously reported in populations characterized by different clinical phenotypes. For example, we did not found correlation between relative abundances of Clostridium species with increased BMI and with higher fasting glucose, findings that were reported in severe obese post-menopausal women (34.5 Kg/m2 as mean BMI) [43], in 70 y-old overweight subjects (28.0 Kg/m2 as mean BMI) but with T2D [44] and in younger and prevalently lean subjects (43-63 y-old and 23.7 Kg/m2 as mean BMI) with diabetes (possibly including type 1 forms) [45].”.

References:

  1. Schnorr, S.L.; Candela, M.; Rampelli, S.; Centanni, M.; Consolandi, C.; Basaglia, G.; Turroni, S.; Biagi, E.; Peano, C.; Severgnini, M.; et al. Gut microbiome of the Hadza hunter-gatherers. Nat. Commun. 2014, 5, doi:10.1038/ncomms4654.

  1. De Filippis, F.; Pellegrini, N.; Vannini, L.; Jeffery, I.B.; La Storia, A.; Laghi, L.; I Serrazanetti, D.; Di Cagno, R.; Ferrocino, I.; Lazzi, C.; et al. High-level adherence to a Mediterranean diet beneficially impacts the gut microbiota and associated metabolome. Gut 2016, 65, 1812–1821, doi:10.1136/gutjnl-2015-309957.

  1. Brahe, L.K.; Le Chatelier, E.; Prifti, E.; Pons, N.; Kennedy, S.; Hansen, T.; Pedersen, O.; Astrup, A.; Ehrlich, S.D.; Larsen, L.H. Specific gut microbiota features and metabolic markers in postmenopausal women with obesity. Nutr. Diabetes 2015, 5, doi:10.1038/nutd.2015.9.

  1. Karlsson, F.H.; Tremaroli, V.; Nookaew, I.; Bergström, G.; Behre, C.J.; Fagerberg, B.; Nielsen, J.; Bäckhed, F. Gut metagenome in European women with normal, impaired and diabetic glucose control. Nature 2013, 498, 99–103, doi:10.1038/nature12198.

  1. Wang, J.; Qin, J.; Li, Y.; Cai, Z.; Li, S.; Zhu, J.; Zhang, F.; Liang, S.; Zhang, W.; Guan, Y.; et al. A metagenome-wide association study of gut microbiota in type 2 diabetes. Nature 2012, 490, 55–60, doi:10.1038/nature11450.

The authors identify some limitations of the study but I consider that an unidentified limitation is the fact that there are significant differences between the average age of the groups with and without carotid atherosclerosis. Because they are younger, they may not have it yet. There may be a confounding factor here. Clarification on this aspect should be added in the limitations of the study.

We agree with the reviewer that age, principal predictor of faster progression of SCA as we reported also in PLIC population (see Ref#4), might act as a confounder for the relation between GM taxonomy and SCA.

Our study was conducted on 86 subjects aged between 40-57 years-old (25.0% of total), 105 subjects aged between 60-70 years-old (30.4% of total), 130 subjects aged between 70-80 years-old (37.7% of total) and 24 subjects aged over 80 years-old (6.9% of total).

(See Table in the attachment).

Based on this distribution, we thus believe that data from larger number of younger groups or longitudinal evaluation on this same cohort are warranted, in order to dissect how much age interacts in the relation between GM composition and SCA progression over time.

As per reviewer’ suggestion, we now implemented this limitation in discussion section (page 15, lines 543-548), as follows:

“…Secondly, subjects with SCA were older in our cohort and we acknowledge that age, the principal predictor of faster SCA progression over time [4], might act as a confounding factor in the relation between GM taxonomy and SCA. However, the majority of subjects in our study were aged between 60 and 80 y-old (Table 1), therefore prompting us to conclude that data from younger cohorts or longitudinal evaluation on this same cohort are needed in order to dissect how much age interacts in the relation between GM composition and SCA progression over time..”.

Reference:

  1. Olmastroni, E.; Baragetti, A.; Casula, M.; Grigore, L.; Pellegatta, F.; Pirillo, A.; Tragni, E.; Catapano, A.L. Multilevel Models to Estimate Carotid Intima-Media Thickness Curves for Individual Cardiovascular Risk Evaluation. Stroke 2019, 50, 1758–1765, doi:10.1161/STROKEAHA.118.024692.

Reviewer 3 Report

This is a very meticulously prepared study, addressing important issue, bringing a novel insights into connections between diets, gut microbiota and subclinical atherosclerosis.

Few remarks:

  • What is the exact definition of SCA used to classify the participants? It is not evident.
  • There seem to be some important baseline differences between the groups with/without SCA (making the conclusions harder) - most importantly the first being much older, with higher waist/hip ratio and more steatosis (implying more metabolic sy), more therapy etc. Have these likely important cofounders been controlled for?
  • What are the possible clinical implications?

Author Response

Dear Reviewer#3,

Thank you very much for your positive and constructive comments. We have now revised the manuscript according to your suggestions and you can find below the responses to each of your comments.

This is a very meticulously prepared study, addressing important issue, bringing a novel insights into connections between diets, gut microbiota and subclinical atherosclerosis.

Few remarks:

  • What is the exact definition of SCA used to classify the participants? It is not evident.

According to reviewer’ suggestion, we now implemented a more detailed description of SCA definition in methods section (page 4, lines 134-163) as follows:

“…SCA was defined by ultrasound-based analysis of bilateral carotid arteries as previously described [27]. In detail, common carotid IMT (one centimeter from the bulb) was measured in longitudinal view, far wall, by high resolution B-mode ultrasound based system (Vivid S5 (GE Healthcareâ, Wauwatosa, WI, USA) connected to linear probe (4.0X13.0 MHz frequency; 14x48 mm footprint, 38 mm field of view)). A mean value for both sides was averaged. “+IMT” was distinguished by “-IMT” in presence of IMT above the 75th percentile of the median IMT for a Caucasian population according to ASE guidelines [29]. SCA was defined when mean IMT was ³ 1.3 mm or in presence of focal atherosclerotic lesions larger than 1.3 mm using a manual caliper in longitudinal view either in far or near wall and over every carotid tract (common, bulb section, bifurcation, internal or external branches). In two scans performed by the same operator in 75 subjects, the mean difference in IMT was 0.005±0.002 mm and the coefficient of variation (CV) was 1.93%. The correlation between two scans was significant (r = 0.96; P < 0.0001). The combination of information from IMT measurement and from presence/absence of SCA allowed to identified four different SCA stages: subjects without intimal thickening and without SCA (“-IMT/-SCA”, n=23); subjects with intimal thickening but without SCA (“+IMT/-SCA”, n=173); subjects without intimal thickening but with SCA (“-IMT/+SCA”, n=121); subjects with both intimal thickening and SCA (“+IMT/+SCA”, n=23).…

For Supplemental Figure 5 (E. coli metagenetic markers associated with advanced SCA progression), further vascular characterization according to validated criteria [30] allowed to distinguish, among the +IMT/+SCA group, subjects with “no advanced SCA” (stenosis <30% and P/S<125 cm/s) vs the “advanced SCA” (stenosis 30% and elevation of the P/S wave in the bilateral internal carotid branches). The advanced SCA was then divided by further characterization identifying: a) SCA causing stenosis between 30 and 50% with P/S <125 cm/s; b) SCA causing stenosis between 50 and 70% and P/S between 125 and 250 cm/s; c) SCA causing stenosis over 70% and P/S over 250 cm/s. We evaluated echolucencies of the atherosclerotic lesions among all subjects from the –IMT/+SCA and from the +IMT/+SCA group, using grey-scale definition and parameters of the QuickScanâ and autoIMTâ software included in the ultrasound machinery (Samsung HM70aâ, Samsungâ, Seoul, South Korea).”.

References:

  1. Baragetti, A.; Pisano, G.; Bertelli, C.; Garlaschelli, K.; Grigore, L.; Fracanzani, A.L.; Fargion, S.; Norata, G.D.; Catapano, A.L. Subclinical atherosclerosis is associated with Epicardial Fat Thickness and hepatic steatosis in the general population. Nutr. Metab. Cardiovasc. Dis. 2016, 26, doi:10.1016/j.numecd.2015.10.013.

  1. Stein, J.H.; Korcarz, C.E.; Hurst, R.T.; Lonn, E.; Kendall, C.B.; Mohler, E.R.; Najjar, S.S.; Rembold, C.M.; Post, W.S.; American Society of Echocardiography Carotid Intima-Media Thickness Task Force Use of Carotid Ultrasound to Identify Subclinical Vascular Disease and Evaluate Cardiovascular Disease Risk: A Consensus Statement from the American Society of Echocardiography Carotid Intima-Media Thickness Task Force Endorsed by the Society for Vascular Medicine. J. Am. Soc. Echocardiogr. 2008, 21, 93–111, doi:10.1016/j.echo.2007.11.011.

  1. Grant, E.G.; Benson, C.B.; Moneta, G.L.; Alexandrov, A. V.; Baker, J.D.; Bluth, E.I.; Carroll, B.A.; Eliasziw, M.; Gocke, J.; Hertzberg, B.S.; et al. Carotid Artery Stenosis: Gray-Scale and Doppler US Diagnosis - Society of Radiologists in Ultrasound Consensus Conference. In Proceedings of the Radiology; 2003; Vol. 229, pp. 340–346.

  • There seem to be some important baseline differences between the groups with/without SCA (making the conclusions harder) - most importantly the first being much older, with higher waist/hip ratio and more steatosis (implying more metabolic sy), more therapy etc. Have these likely important cofounders been controlled for?

We thank the reviewer for raising this important point.

It should be noted that significant differences we observed in microbial composition comparing beta-diversity measures via Principal Coordinate Analysis (PCoA) still take into account potential confounders, since they are based on multivariate models, associated with data-reduction techniques that produce a set of uncorrelated (orthogonal) axes to summarize the variability in the data set.

This aspect has been now acknowledged in the limitations sections (page 15, lines 581-585):

“…the cross-sectional design of this investigation cannot rule out the burden of CVRFs prior to the examination, although significant differences we observed in microbial composition comparing beta-diversity measures via Principal Coordinate Analysis (PCoA) still take into account potential confounders, since they are based on multivariate models, associated with data-reduction techniques that produce a set of uncorrelated (orthogonal) axes to summarize the variability in the data set.”.

Nevertheless, we now evaluated here below, one by one, the potential confound effect of each single parameter that was significantly different between the groups of subjects with and without SCA.

Age:

Our study was conducted on 86 subjects aged between 40-57 years-old (25.0% of total), 105 subjects aged between 60-70 years-old (30.4% of total), 130 subjects aged between 70-80 years-old (37.7% of total) and 24 subjects aged over 80 years-old (6.9% of total).

(See Table in the attachment).

Based on this distribution, we thus believe that data from larger number of younger groups or longitudinal evaluation on this same cohort are warranted, in order to dissect how much age interacts in the relation between GM composition and SCA progression over time.

We now dissected this point in discussion section (page 15, lines 591-596), as follows:

“…Secondly, subjects with SCA were older in our cohort and we acknowledge that age, the principal predictor of faster SCA progression over time [4], might act as a confounding factor in the relation between GM taxonomy and SCA. However, the majority of subjects in our study were aged between 60 and 80 y-old (Table 1), therefore prompting us to conclude that data from younger cohorts or longitudinal evaluation on this same cohort are needed in order to dissect how much age interacts in the relation between GM composition and SCA progression over time..”.

Reference:

  1. Olmastroni, E.; Baragetti, A.; Casula, M.; Grigore, L.; Pellegatta, F.; Pirillo, A.; Tragni, E.; Catapano, A.L. Multilevel Models to Estimate Carotid Intima-Media Thickness Curves for Individual Cardiovascular Risk Evaluation. Stroke 2019, 50, 1758–1765, doi:10.1161/STROKEAHA.118.024692.

Waist-hip ratio and hepatic steatosis:

Although we excluded from this analysis subjects who presented with T2D or were diagnosed with MetS during previous outpatient’s examination of the PLIC study), SCA was associated with increased waist-hip ratio and higher prevalence of hepatic steatosis (ultrasound-defined).

However, we did not find differences in either alpha (p>0.578 on all metrics) or beta diversities (p=1 for both unweighted and weighted Unifrac distances, respectively).

(See the attachment for representative plots of alpha and beta diversities).

Moreover, when repeating the analyses including only subjects without hepatic steatosis, we further confirmed differences in alpha and beta diversities between subjects with SCA versus those without SCA. Although possibly excluding these two markers as confounders, we cannot rule out that the poor number of subjects with hepatic steatosis (n=31) might underrepresents the actual effect of this parameter on GM composition.

Based on this finding, we implemented this point in discussion section (page 14, lines 477-484), as follows:

“…Also, hepatic steatosis (which was only ultrasound-determined and poorly prevalent in our cohort) as well as plasma hs-CRP, produced by hepatocytes marking low-grade inflammation, did not correlate either with GM diversity (while recently reported in morbidly obese patients [46] and in liver steatosis [47]), or with reduced relative abundance of specific bacterial species (e.g.: Akkermansia muciniphila (less abundant in insulin resistance [48] and in obese individuals [49]). Although these data might exclude an impact of a gut-liver connection during early stages of SCA, in depth evaluation on larger cohorts with more advanced stages of liver disease should are warranted..”.

References:

  1. Vieira-Silva, S.; Falony, G.; Belda, E.; Nielsen, T.; Aron-Wisnewsky, J.; Chakaroun, R.; Forslund, S.K.; Assmann, K.; Valles-Colomer, M.; Nguyen, T.T.D.; et al. Statin therapy is associated with lower prevalence of gut microbiota dysbiosis. Nature 2020, 1–6, doi:10.1038/s41586-020-2269-x.

  1. Aron-Wisnewsky, J.; Gaborit, B.; Dutour, A.; Clement, K. Gut microbiota and non-alcoholic fatty liver disease: New insights. Clin. Microbiol. Infect. 2013, 19, 338–348.

  1. Schneeberger, M.; Everard, A.; Gómez-Valadés, A.G.; Matamoros, S.; Ramírez, S.; Delzenne, N.M.; Gomis, R.; Claret, M.; Cani, P.D. Akkermansia muciniphila inversely correlates with the onset of inflammation, altered adipose tissue metabolism and metabolic disorders during obesity in mice. Sci. Rep. 2015, 5, doi:10.1038/srep16643.

  1. Dao, M.C.; Everard, A.; Aron-Wisnewsky, J.; Sokolovska, N.; Prifti, E.; Verger, E.O.; Kayser, B.D.; Levenez, F.; Chilloux, J.; Hoyles, L.; et al. Akkermansia muciniphila and improved metabolic health during a dietary intervention in obesity: Relationship with gut microbiome richness and ecology. Gut 2016, 65, 426–436, doi:10.1136/gutjnl-2014-308778.

Pharmacological therapies (anti-hypertensive and lipid lowering drugs):

Subjects with advanced SCA stages were more on anti-hypertensive drugs.

However, we did not found significant differences in either alpha (p>0.7 for all considered metrics) or beta diversities between subjects on anti-hypertensive as compared to those without anti-hypertensive (p=0.531 and p=0.479 for unweighted and weighted Unifrac distances, respectively).

(See the attachment for representative plots of alpha and beta diversities).

We performed similar analysis for lipid lowering drugs (which were more prescribed in subjects with SCA, explaining their lower cholesterol and triglycerides levels as compared to those without SCA). Similarly, we did not find differences in either alpha (p=1 on all metrics) or beta diversities (p= 0.51 and p= 0.186 for unweighted and weighted Unifrac distances, respectively) between subjects on lipid lowering drugs as compared to those without lipid lowering drugs.

(See the attachment for representative plots of alpha and beta diversities).

Based on these findings, we now implemented this point in discussion section (page 14, lines 580-590), as follows:

“.... Also the effect of pharmacological interventions over time (especially considering anti-hypertensive and lipid lowering drugs) cannot be excluded; in fact, we did not find differences in the GM alpha and beta diversities comparing subjects with pharmacological therapy and those without any treatments but we do not have precise information nor about their effect on GM composition neither on the bioavailability of these drugs (an aspect that could be particularly of interest for statins [23,63]).…”.

References:

  1. Jie, Z.; Xia, H.; Zhong, S.-L.; Feng, Q.; Li, S.; Liang, S.; Zhong, H.; Liu, Z.; Gao, Y.; Zhao, H.; et al. The gut microbiome in atherosclerotic cardiovascular disease. Nat. Commun. 2017, 8, 845, doi:10.1038/s41467-017-00900-1.

  1. Tuteja, S.; Ferguson, J.F. Gut Microbiome and Response to Cardiovascular Drugs. Circ. Genomic Precis. Med. 2019, 12, 421–429.

  • What are the possible clinical implications?

We studied connections between taxonomic and metagenomic changes in subjects with different stages of SCA (but in primary prevention for ACVD) and self-reporting dietary data to derive the individual exposure to diet.

We cannot rule out whether and how dietary exposure differently affects GM composition in presence or absence of SCA but the immediate clinical implication is to address, by future longitudinal dietary interventional trials, if this relation implies causality. In the long-term, this approach might provide a pioneering tentative of personalized prevention against ACVD.

On the other way round, these data also call for improvements in the engagement of patients to improve the communication with dieticians and clinicians and to increase the value of healthy diet still in primary prevention (e.g: smartphone health apps or other health mobile technologies used, in a telemedicine-based approach).

These aspects have been now added over the text:

  1. a) in introduction section (page 2-3, lines 87-97):

“…This scientific question is of actual clinical concern given the continuous changes in dietary sources nowadays among societies [6]. However, methodological criticisms affect GM composition analysis [24], the design of interventional dietary trials is up to date scarce, small-sized trials gave contrasting data about the effect of dietary intervention on changes of GM composition and subsequent effect on markers of ACVD risk [25]. This sets the stage for an immediate clinical value, since the clustering of taxonomic and metagenomic signatures with individual dietary lifestyle might represent a pioneering approach of primary prevention, identifying patients among the population at increased risk of future occurrence of CVD. We here address the relation between functional metagenomic signatures and individual exposure to diet during subclinical manifestation of CVD, studying people from a general population-based study, in primary prevention, with low prevalence of CVRFs and characterized by different stages of SCA”.

References:

  1. Yusuf, S.; Joseph, P.; Rangarajan, S.; Islam, S.; Mente, A.; Hystad, P.; Brauer, M.; Kutty, V.R.; Gupta, R.; Wielgosz, A.; et al. Modifiable risk factors, cardiovascular disease, and mortality in 155 722 individuals from 21 high-income, middle-income, and low-income countries (PURE): a prospective cohort study. Lancet (London, England) 2019, doi:10.1016/S0140-6736(19)32008-2.
  2. Swann, J.R.; Rajilic-Stojanovic, M.; Salonen, A.; Sakwinska, O.; Gill, C.; Meynier, A.; Fança-Berthon, P.; Schelkle, B.; Segata, N.; Shortt, C.; et al. Considerations for the design and conduct of human gut microbiota intervention studies relating to foods. Eur. J. Nutr. 2020, 59, 3347–3368.
  3. Gerdes, V.; Gueimonde, M.; Pajunen, L.; Nieuwdorp, M.; Laitinen, K. How strong is the evidence that gut microbiota composition can be influenced by lifestyle interventions in a cardio-protective way? Atherosclerosis 2020, 311, 124–142.

  1. b) in conclusions section (page 15, lines 523-529):

“…Firstly, this epidemiological study, gathering self-reported dietary data, does not allow to unveil the actual relation of causality. These limitations pave the road to dissect this aspect in the near future, a perspective that might be pursued: i) by clustering larger number of subjects one the basis of their exposure to different dietary patterns (daily collected using smartphone health apps or other health mobile technologies to improve self-reporting) or ii) by longitudinally evaluating the actual effect of a single dietary pattern/habit on changes of GM composition/functionality in subjects with different SCA stages.”.

  1. c) in conclusions section (page 16, lines 560-563):

“.... This cross-sectional data aim at setting the stage for future longitudinal studies and dietary interventions, testing if personalized modifications in dietary habits over time could affect GM composition contributing to the prevent of the onset of CVRF and the clinical manifestation of ACVD.”.

Round 2

Reviewer 1 Report

Thank you for your detailed point-to-point response and for addressing all my comments. I have no further comments or concerns.